# Ammonia for post-healing of formamidinium-based Perovskite films

Zhipeng Li[1,2], Xiao Wang[1], Zaiwei Wang[1], Zhipeng Shao[1], Lianzheng Hao[1,2], Yi Rao[1,2], Chen Chen[1], Dachang Liu[1,2], Qiangqiang Zhao[1,3], Xiuhong Sun[1,2], Caiyun Gao[1], Bingqian Zhang[1], Xianzhao Wang[1,2], Li Wang[3] ✉, Guanglei Cui[1,4] ✉ & Shuping Pang[1] ✉

Solvents employed for perovskite film fabrication not only play important roles in dissolving the precursors but also participate in crystallization process. High boiling point aprotic solvents with O-donor ligands have been extensively studied, but the formation of a highly uniform halide perovskite film still requires the participation of additives or an additional step to accelerate the nucleation rate. The volatile aliphatic methylamine with both coordinating ligands and hydrogen protons as solvent or post-healing gas facilitates the process of methylamine-based perovskite films with high crystallinity, few defects, and easy large-scale fabrication as well. However, the attempt in formamidinium-containing perovskites is challenged heretofore. Here, we reveal that the degradation of formamidinium-containing perovskites in aliphatic amines environment results from the transimination reaction of formamidinium cation and aliphatic amines along with the formation of ammonia. Based on this mechanism, ammonia is selected as a post-healing gas for a highly uniform, compact formamidinium-based perovskite films. In particular, low temperature is proved to be crucial to enable formamidinium-based perovskite materials to absorb enough ammonia molecules and form a liquid intermediate state which is the key to eliminating voids in raw films. As a result, the champion perovskite solar cell based on ammonia post-healing achieves a power conversion efficiency of 23.21% with excellent reproducibility. Especially the module power conversion efficiency with 14 cm² active area is over 20%. This ammonia post-healing treatment potentially makes it easier to upscale fabrication of highly efficient formamidinium-based devices.

Halide perovskite materials with the general formula ABX₃, where A refers to a monovalent cation such as methylammonium ($CH_3NH_3^+$, $MA^+$), formamidinium ($HC(NH_2)_2^+$, $FA^+$), $Cs^+$, B represents a divalent cation such as $Pb^{2+}$, $Sn^{2+}$, $Ge^{2+}$, and X represents a halide ion such as $I^-$, $Br^-$, $Cl^-$, have emerged as a class of promising light-harvesting materials in photovoltaics since 2009[1]. The highest certified power conversion efficiency (PCE) of perovskite solar cells (PSCs) has so far been up to 25.7%[2], which is comparable to that of widely commercialized Si-based solar cells. The crystal structure of halide perovskites can be regarded as a $[MX_6]^{4-}$ octahedron in a three-dimensional (3D) space with a

[1]Qingdao Institute of Bioenergy and Bioprocess Technology, Chinese Academy of Sciences, Qingdao 266101, P. R. China. [2]Center of Materials Science and Optoelectronics Engineering, University of Chinese Academy of Sciences, Beijing 100049, P. R. China. [3]College of Materials Science and Engineering, Qingdao University of Science and Technology, Qingdao 266042, P. R. China. [4]School of Future Technology, University of Chinese Academy of Sciences, Beijing 100049, P. R. China. ✉e-mail: liwang718@qust.edu.cn; cuigl@qibebt.ac.cn; pangsp@qibebt.ac.cn

common apex angle connecting. The A site ions fill up the gap of the octahedral and the structure is stabilized by Van der Waals forces[3]. These special structural characteristics provide a variety of solution processing methods to prepare perovskite films[4–6].

The high boiling point aprotic *N,N*-dimethylformamide (DMF), *γ*-butyrolactone (GBL), dimethyl sulfoxide (DMSO) and *N,N*-dimethylacetamide (DMA) are the common solvents employed to dissolve the perovskite precursors with the formation of solvated iodoplumbate complexes because of the strong coordination between O-donor ligands (OLs) and Pb(II)[4,7–9]. The Lewis basicity of solvents is thought to correlate with the "coordinating ability" with lead halide salts, which can be predicted by their Gutmann's Donor Number ($D_N$) with a trend of DMSO > DMA > DMF > GBL[10,11]. High $D_N$ solvents with strong coordination with the Pb(II) center also result in the formation of intermediate phase OL-PbI$_2$-RAI (RA refers to MA or FA, etc.) prior to perovskite phase during the film fabrication process. The anisotropic growth nature of the intermediate phase leads to rough films with one-dimensional (1D) fiber-like structures and a large proportion of void area[12]. In this regard, stronger coordinating additives (such as thiourea[13] and pyridine[14]) and/or the fast nucleation strategies (such as anti-solvent extraction, gas-quenching, and vacuum-assisted drying) have been introduced to regulate the growth process of the intermediate phase to achieve highly uniform perovskite films with good crystallinity[7,15,16].

Among them, methylamine (MA$^0$) featured with the presence of hydrogen bonding and low boiling point has become an impressive coordination agent for the MAPbI$_3$ material. In this case, the formed intermediate phase is a metastable (PbI$_2$-MAI)·xMA$^0$ complex, instead of simple coordinate bond-dominated OL-PbI$_2$-MAI adducts. Easy formation of highly uniform (PbI$_2$-MAI)·xMA$^0$ films and following conversion to highly uniform perovskite films are closely related to the self-leveling behavior of (PbI$_2$-MAI)·xMA$^0$ liquid intermediate phase and the ultrafast evaporation of MA$^0$ gas. MA$^0$ employed as a post-healing gas to eliminate the voids in the MAPbI$_3$ perovskite film is firstly reported by Zhou et al. in 2015[17], and the adoption of MA$^0$ as a volatile solvent system has become a commercially viable technology for MAPbI$_3$ devices with excellent device reproducibility[18–23]. In comparison with MAPbI$_3$, α-FAPbI$_3$ has higher theoretical efficiency and thermal stability. The undesired phase transition to non-perovskite phase δ-FAPbI$_3$ has been well dissolved by alloying a small amount of Cs$^+$, MA$^+$ to tailor the tolerance factor[24–29]. We have attempted to introduce MA$^0$ gas-related methods for the fabrication of FA-containing perovskite layers, unfortunately, resulting in degradation of the 3D perovskite phase[18]. The underlying reason for the irreversible transformation of FAPbI$_3$ with MA$^0$ treatment is still not fully clear, let alone make efforts to solve this problem.

Here, we have systematically studied the underlying chemical reactions between aliphatic amines/formamidine (FA$^0$) gases and FAI salt, and elucidated the addition-elimination reaction between amine compound and the imine band of FA$^0$ with the formation of ammonia (NH$_3$), also named transimination reaction. Based on this mechanism, NH$_3$ is selected as a post-healing gas to avoid the degradation of the FA-based perovskite phase during the post treatment. It is demonstrated that decreasing processing temperature is crucial for FA-based perovskite layer to enhance the absorption of NH$_3$ molecules, leading to the formation of a desired liquid intermediate state. Self-leveling behavior of the liquid intermediate state can quickly heal the voids in the rough perovskite film and finally form a highly uniform and compact film with the evaporation of NH$_3$. At last, the PCE of FA-based PSC based on this NH$_3$ post-healing strategy is more than 23% with a certified PCE of 22.22%. Especially, the PSC module achieves a PCE of 20.61% which is comparable with the highest reports in PSC modules. These results demonstrate the large advantage of the NH$_3$ gas post-healing technology in upscaling fabrication of highly efficient PSCs.

# Results

## The degradation evidence of formamidinium-containing perovskites in aliphatic amines environment

Nuclear magnetic resonance (NMR) spectroscopy is a particularly practical tool for quantifying the relative amounts of the organic cations and related chemical reactions in the perovskite precursor[30–33]. We started with the study of the underlying chemical reactions between FA$^0$/R-NH$_2$(RA$^0$) gases and FAI salt (Table 1). The gases involved include FA$^0$, MA$^0$, ethylamine (EA$^0$), *n*-propylamine (PA$^0$), *n*-butylamine (BA$^0$), and NH$_3$. The detailed preparation procedures of the relevant gases are provided in the methods section.

The synthesis of FA$^0$ is tricky because of its instability. Mixed FACl and NaOH powders are expected to form FA$^0$ by the neutralization as shown in Supplementary Scheme 1a. To study their product, deuterated DMSO (DMSO-d$_6$) solvent is used to collect released gas from the mixed powders at room temperature (RT) by a simple equipment (Supplementary Fig. 1a) for $^1$H nuclear magnetic resonance ($^1$H NMR) measurement. Only NH$_3$ signal is detected in the $^1$H NMR (Table 1 (No. 1) and Supplementary Fig. 4). We think that the high boiling point of FA$^0$ likely makes FA$^0$ molecules difficult to diffuse into DMSO-d$_6$ solvent. The formation of NH$_3$ is indicative of the existence of some chemical reactions in the process of gas preparation (Supplementary Fig. 2). When the temperature of the reactor and the gas pipeline is increased to 150 °C and DMSO-d$_6$ solvent is still kept at RT, four main compounds are detected: *s*-triazine, formamide, FA$^0$, and NH$_3$ (Table 1 (No. 2) and Supplementary Fig. 5). The formation of *s*-triazine is because of the transimination reaction of three FA$^0$ molecules along with the formation of NH$_3$ (Supplementary Fig. 2b)[30,31]. The *s*-triazine has also been detected when the FAI powder is heated at 150 °C[32]. Formamide is attributed to the hydrolyzation product of FA$^0$ (Supplementary Fig. 2c) in the presence of H$_2$O which is generated from the neutralization between FACl and NaOH. Considering the long-time interval from gas collection to $^1$H NMR measurement, we then employ acetic acid (HOAc) to capture FA$^0$ so as to limit these chemical reactions in DMSO-d$_6$ solvent by the quick formation of FAAc. As expected, $^1$H NMR spectrum verifies that the reaction product at 150 °C only exists FAAc rather than *s*-triazine, formamide, or NH$_4$Ac (Table 1 (No. 3) and Supplementary Fig. 6). The above results strongly indicate the

**Table 1 | The synthesis of formamidine (FA$^0$) and chemical reactions between FA$^0$/amines/NH$_3$ gases and FAI salt**

| No. | Gas generation | Gas collection | Product[a] |
|---|---|---|---|
| 1 | FACl, NaOH, RT | DMSO-d$_6$ | NH$_3$ |
| 2 | FACl, NaOH, 150 °C | DMSO-d$_6$ | *s*-triazine, formamide, FA, NH$_3$ |
| 3 | FACl, NaOH, 150 °C | HOAc | FAAc[b] |
| 4 | FACl, NaOH, 150 °C | FAI powder | FAI, formamide[b] |
| 5 | MA in EtOH, 60 °C | FAI powder | DMFAI[b] |
| 6 | EACl, NaOH, 60 °C | FAI powder | DEFAI[b] |
| 7 | PA, 60 °C | FAI powder | DPFAI[b] |
| 8 | BA, 60 C | FAI powder | DBFAI[b] |
| 9 | NH$_3$·H$_2$O, 60 °C | FAI powder | NH$_4$I[b] |
| 10 | MA in H$_2$O, 60 °C | FAI powder | MAI[b] |
| 11 | EA in H$_2$O, 60 °C | FAI powder | EAI[b] |
| 12 | H$_2$O, 60 °C | FAI powder | FAI[b] |
| 13 | NH$_3$, RT | FAI powder | FAI[b] |

[a] The main components are listed.
[b] The samples are treated with vacuum to remove low boiling point components before $^1$H NMR measurement. FACl is formamidine hydrochloride, FA is formamidine, HOAc is acetic acid, FAAc is formamidinium acetic, FAI is formamidinium iodide, MA is methylamine, EtOH is ethyl alcohol, DMFAI is *N,N'*-dimethyl formamidinium iodide, EACl is ethylamine hydrochloride, DEFAI is *N,N'*-diethyl formamidinium iodide, PA is *n*-Propylamine, DPFAI is *N,N'*- dipropyl formamidinium iodide, BA is *n*-Butylamine, DBFAI is *N,N'*- dibutyl formamidinium iodide. MAI is methylaminium iodide, and EAI is ethylamine hydroiodide.

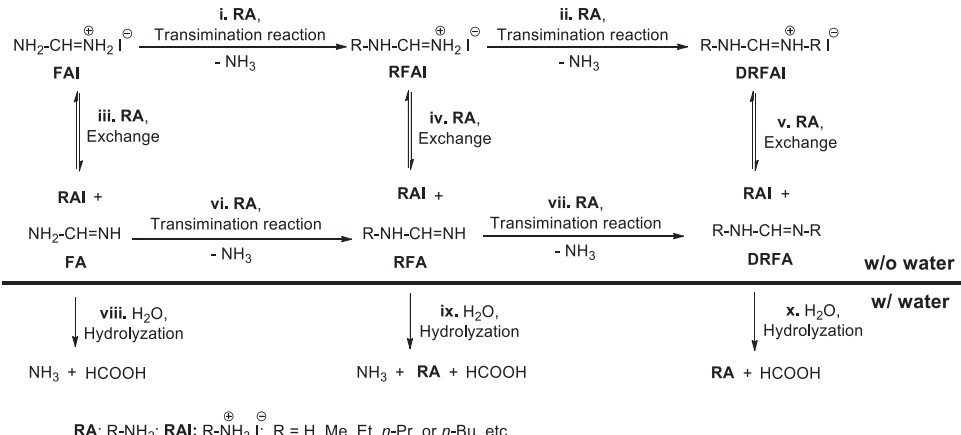

**Fig. 1 | Chemical reactions related to FAI.** The transimination reactions and the ion exchange reactions between FAI and $RA^0$ molecules (R refers to H, Me, Et, $n$-Pr, or $n$-Bu etc.), and the hydrolysis reactions of $FA^0$, $RFA^0$, and $DRFA^0$.

formation and instability of $FA^0$. When FAI powder is exposed to $FA^0$, there is hardly any change in chemical compositions except the formation of a tiny amount of formamide as the presence of $H_2O$ (Table 1 (No. 4), Supplementary Fig. 7). These spontaneous reactions and instability of $FA^0$ seriously limit its practical application in post-healing FA-based perovskite films.

In comparison with $FA^0$, amines are more stable. When FAI powder is treated with $MA^0$, $EA^0$, $PA^0$, or $BA^0$, the powder quickly transforms into a liquid state. After the liquid is held at 60 °C for 6 h and then vacuumed (Supplementary Fig. 1b), it is unexpected that there are no FAI signals in $^1H$ NMR spectrum of the final powder samples (Table 1 (No. 5–8), Supplementary Figs. 8–11).

Taking $MA^0$ as an example (Table 1 (No. 5)), we find that the main signals in $^1H$ NMR spectrum belong to $N,N'$-dimethyl formamidinium iodide (DMFAI) (Supplementary Figs. 3 and 8)[33]. The related chemical reactions are illustrated in Fig. 1 (i, ii). The lone-pair electrons of N atom in nucleophilic $MA^0$ molecule can attack electrophilic imine bond in FAI, which leads to the formation of $N$-methyl formamidinium iodide (MFAI) by a transimination process (Fig. 1(i), Supplementary Fig. 3)[34]. The imine bond in the formed MFAI can carry out the second transimination reaction with $MA^0$ to form DMFAI (Fig. 1(ii), Supplementary Fig. 3). With excessive $MA^0$ and holding for enough time, FAI can fully transform into DMFAI. Similarly, when FAI powders are treated with $EA^0$, $PA^0$ and $BA^0$ gases, the transimination reaction also occur with the formation of DEFAI, DPFAI and DBFAI, respectively (Table 1 (No. 6–8), Supplementary Figs. 9–11, Fig. 1(i, ii)).

When FAI powder is treated by $RA^0$ (R is referred to H, Me, Et etc.) gas with some $H_2O$ vapor evaporated from their aqueous solution (Supplementary Fig. 1c), the final product is RAI rather than FAI, RFAI or DRFAI (Table 1 (No. 9–11), Supplementary Figs. 12–14). Considering there is no reaction between FAI salt and $H_2O$ (Table 1 (No. 12), Supplementary Fig. 15), the formation of RAI mentioned above is due to the reversible ion exchange reactions between FAI, RFAI or DRFAI and $RA^0$ with the formation of RAI and $FA^0$, $RFA^0$, or $DRFA^0$. The hydrolysis reactions of $FA^0$, $RFA^0$, and $DRFA^0$ lead to the formation of volatile HCOOH and $RA^0$, which in turn promotes the ion exchange reactions towards the formation of RAI (Fig. 1 (iii–x))[35].

Supplementary Fig. 16 shows the optical photos and XRD patterns of $\alpha$-FAPbI$_3$ films before and after $MA^0$ post-healing treatment. $\delta$-FAPbI$_3$, MFAPbI$_3$, and DMFAPbI$_3$ films are also measured for comparison. The formation of the non-perovskite phase after $MA^0$ post-healing the $\alpha$-FAPbI$_3$ film is because of the conversion from $FA^+$ to $MFA^+$ and $DMFA^+$. This is why aliphatic amines cannot be employed as solvents or post-healing gases for the fabrication of FA-containing perovskite materials. Inspired by the above transimination reaction, $NH_3$ is selected to avoid the change of composition after the gas

post-healing treatment. The $^1H$ NMR spectrum well verifies no composition change of FAI powder after being treated with $NH_3$ gas (Table 1 (No. 13), Supplementary Fig. 17).

## Feasibility analysis of ammonia as the healing gas

The reversible absorption-desorption process of gas is crucial for the post-healing strategy. MAI and FAI samples are weighted during $MA^0$/$NH_3$ absorption and desorption as shown in Fig. 2a. The whole process can be divided into gas absorption at RT (Stage I), desorption at RT (Stage II), and desorption at 80 °C (Stage III). When exposed to $MA^0$, MAI quickly absorbs $MA^0$ and reaches a saturated state with the $x$ value of about 2.5 in MAI·$x$MA$^0$ (the $x$ value is very sensitive to temperature and pressure). The liquid MAI·$x$MA$^0$ complex removed from $MA^0$ atmosphere desorbs $MA^0$ gas spontaneously until it transforms into a relative stable semi-solid-state with the $x$ value of about 1.1, which is denoted as stage II. After further heating at 80 °C for 10 min (Stage III), the sample returns to a white powder with the same weight as the initial MAI, which means the complete reversibility of absorption-desorption process in MAI-$MA^0$ pair. In FAI-$MA^0$ pair, the Stage I and Stage II are similar to these of MAI-$MA^0$ pair, but the weight of the final powder in the Stage III cannot return back to the initial value of FAI because of the formation of MFAI and DMFAI as presented in Fig. 2a.

FAI powder exposed to $NH_3$ gas atmosphere rapidly achieves a liquid phase FAI·$x$NH$_3$ (Supplementary Fig. 18) with the x value of ~3.0 (Stage I). The liquid phase transforms into an ice-like transparent solid-state FAI·$x$NH$_3$ with $x$ ~ 1.3 as $NH_3$ desorption in Stage II and the XRD pattern of the ice-like complex is shown in Supplementary Fig. 17. The following thermal annealing at 80 °C fully removes the absorbed $NH_3$ molecules and leads to the formation of a white powder with the same weight as the initial FAI (Stage III). This phenomenon, similar to that of MAI-$MA^0$ pair, preliminarily indicates the feasibility of the $NH_3$ as a post-healing gas for FA-based perovskite films.

To make a profound study, the chemical reaction and intermolecular interactions between FAPbI$_3$ precursors and $NH_3$ molecules, the solution $^{15}N$ NMR spectra are measured as shown in Fig. 2b and Supplementary Figs. 19–24. The measured isotopically labeled $^{15}N$ samples are $^{15}NH_3$, FAI treated with $^{15}NH_3$, and vacuumed (FAI($^{15}N$)), FAI absorbing $^{15}NH_3$ molecules (FAI·$x^{15}NH_3$), PbI$_2$ absorbing $^{15}NH_3$ molecules (PbI$_2$·$x^{15}NH_3$), and FAPbI$_3$ absorbing $^{15}NH_3$ molecules (FAPbI$_3$·$x^{15}NH_3$). The $^{15}N$ chemical shift of $^{15}NH_3$ in DMSO-d$_6$ is −377.46 ppm. The $^{15}N$ NMR spectrum of FAI after treatment with $^{15}NH_3$ (FAI ($^{15}N$)) shows the $^{15}N$ signal at −356.33 ppm, which indicates the existence of transimination reaction between $NH_3$ and FAI (Supplementary Fig. 3). The reaction process between the $NH_3$ and FAI is similar to the transimination reaction involving the aliphatic amine. The lone-pair electrons of N atom in $NH_3$ can attack electrophilic imine bond in FAI,

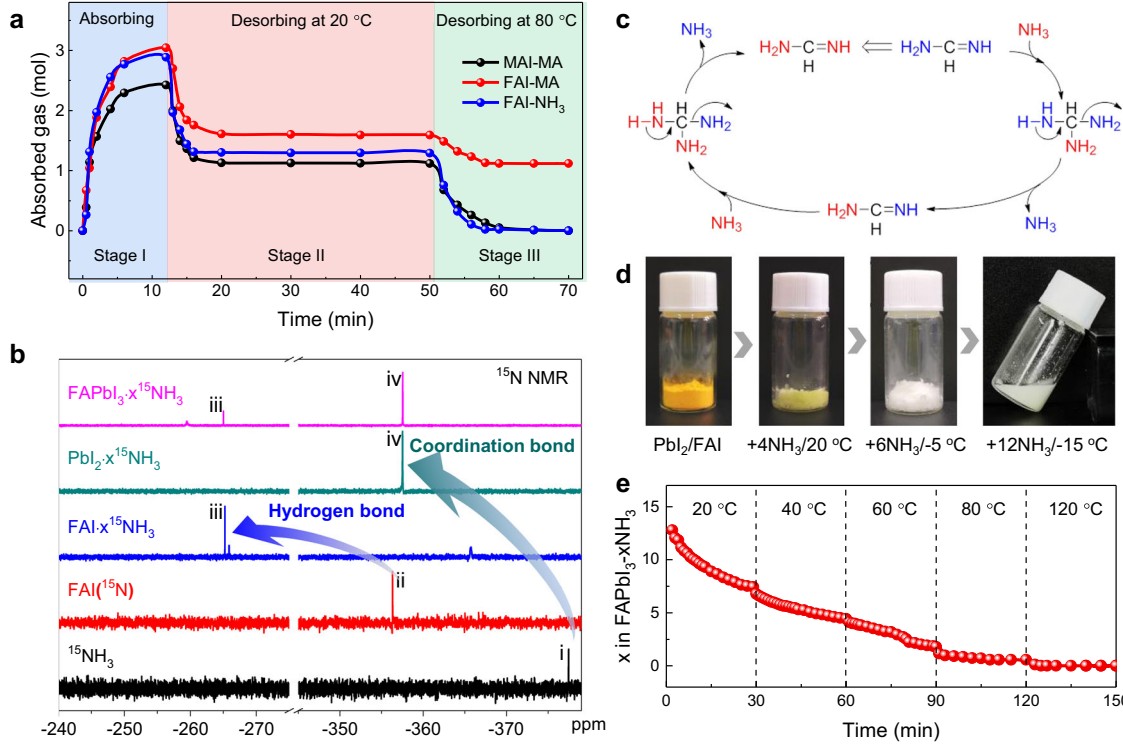

**Fig. 2 | The interaction of $NH_3$ with FA-based perovskite or precursors.**
**a** Absorption and desorption behavior of MAI·MA[0], FAI·MA[0], and FAI·NH₃. **b** The solution $^{15}N$ NMR spectra of $^{15}NH_3$, FAI($^{15}N$), FAI·x$^{15}NH_3$, PbI₂·x$^{15}NH_3$, and FAPbI₃·x$^{15}NH_3$. Signals: i, $^{15}N$ signal in $^{15}NH_3$; ii, $^{15}N$ signal in FAI ($^{15}N$); iii, $^{15}N$ signal in FAI hydrogen bonded with $^{15}NH_3$; iv, $^{15}N$ signal of $^{15}NH_3$ coordinated with Pb(II).

**c** Reaction mechanism of transimination reactions between $NH_3$ and FAI. **d** The photographs of PbI₂/FAI powder in $NH_3$ atmosphere under different temperatures. **e** The calculated $x$ value of FAPbI₃·x$NH_3$ complex in an open condition under different temperatures.

leading to the exchange of $^{15}N$ in $NH_3$ and $^{14}N$ in FAI and formation of FAI($^{15}N$), as shown in Fig. 2c. Besides, the $^{15}N$ chemical shift of FAI in FAI·x$^{15}NH_3$ is at −265.22 ppm, which moves towards the low field about 91 ppm in comparison with FAI($^{15}N$). It suggests the presence of hydrogen bond interaction between $NH_3$ and FAI. Furthermore, the $^{15}N$ chemical shift of PbI₂·x$^{15}NH_3$ at −357.51 ppm moves towards the low field about 20 ppm compared to that of $^{15}NH_3$, which is due to the coordination of $^{15}NH_3$ to Pb(II). The $^{15}N$ chemical shifts of FAPbI₃·x$^{15}NH_3$ at −265.02 ppm and −357.53 ppm correspond to that of FA$^+$($^{15}N$) with hydrogen bond interaction with $NH_3$ and that of $^{15}NH_3$ with coordination bond interaction with Pb(II), respectively. The coexistence of hydrogen bond and coordination bond in the intermediate phase FAPbI₃·x$^{15}NH_3$ is also supported by the solid-state $^{15}N$ NMR result in Supplementary Fig. 25.

$NH_3$, employed as a post-healing gas, should also have good solubility for the FA-based perovskite precursors, including PbI₂ and FAI powders. The color of mixed PbI₂/FAI powder exposed in $NH_3$ gas atmosphere turns into light yellow from the yellow of PbI₂ at 20 °C[36]. The mass difference before and after absorption shows that the absorbed $NH_3$ molecule number per FAPbI₃ is ~4. While FAPbI₃·4$NH_3$ is still a solid-state rather than an expected liquid state, which is very different from liquid MAPbI₃·xMA with x of only 3[17]. One possible reason is the much smaller size of $NH_3$ than MA[0]. We have found that the absorbed amount of gas is highly dependent on processing temperature[23]. Lowering the processing temperature to −5 °C leads to the formation of a white powder with $x$ value of ~6. At −15 °C, a flowable slurry with the $x$ value of ~12 is formed, as shown in Fig. 2d.

The kinetic process of $NH_3$ desorption from the flowable FAPbI₃·x$NH_3$ complex is measured by monitoring its weight change with temperature increased step-by-step from 20 °C to 120 °C under an open condition. The absorbing $NH_3$ number per FAPbI₃ as the function of time illustrated in Fig. 2e shows that the weight of FAPbI₃·x$NH_3$ slurry drops rapidly at the initial stage and then decreases slowly in 30 min at 20 °C. Then the sample was annealed at 40 °C for 30 min and 60 °C for 30 min showing continuously slow weight loss by weighing at regular intervals. At 80 °C, the sample quickly reaches a relatively stable state then almost no weight loss in the maintained 30 min and the $NH_3$ molecular number per FAPbI₃ is about 1.0. Finally, the $x$ value quickly decreases to 0 when the sample is annealed at 120 C.

## Preparation and characterization of thin films

Then the $NH_3$ post-healing strategy is employed to prepare FA-based perovskite films based on a homemade chamber with a semiconductor chilling plate controlling chamber temperature (Supplementary Fig. 26). As shown in Supplementary Fig. 27a, b, the raw FAPbI₃ film with voids by the traditional one-step spin-coating method can transform into a uniform film after $NH_3$ post-healing at −15 °C. The $NH_3$ post-healing FAPbI₃ is denoted as $NH_3$-FAPbI₃. The XRD spectra show that the $NH_3$-FAPbI₃ film has better crystallinity and less undesired δ phase than the raw film (Supplementary Fig. 27c). This leads to the increasing PCE of the $NH_3$-FAPbI₃ device compared to the raw-FAPbI₃ device (Supplementary Fig. 27d). While the PCE of $NH_3$-FAPbI₃ device is still limited due to the existence of undesired δ-phase in the perovskite film. To prepare highly efficient PSCs, cesium (Cs) doped FAPbI₃ material system FA₀.₉Cs₀.₁PbI₃ (FACsPbI₃) is selected. The $NH_3$ post-healing process of FACsPbI₃ film is schematically illustrated in Fig. 3a. Due to the thin thickness of perovskite films, the $NH_3$ gas absorption and desorption are much easier and faster than those of powder samples. The SEM images show that the raw FACsPbI₃ film could be well healed by the $NH_3$ post-healing method in the temperature range from −15 °C to 0 °C, and −15 °C was used for $NH_3$ gas post-healing in the following study (Supplementary Fig. 28).

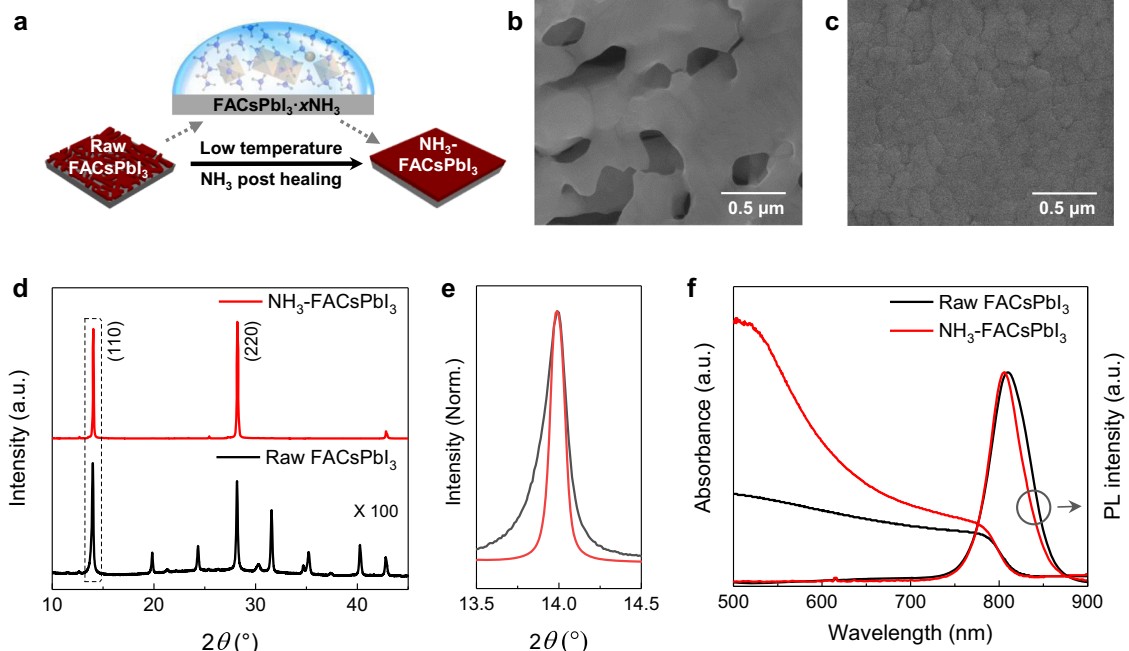

**Fig. 3 | Properties of perovskite Films. a** Schematic illustration of $NH_3$ post-healing $FACsPbI_3$ perovskite thin film. **b, c** Scanning electron microscope (SEM) images of (b) raw $FACsPbI_3$ and (c)$NH_3$-$FACsPbI_3$ perovskite films. **d, e** X-ray diffraction (XRD) patterns, **f** Ultraviolet-visible (UV-Vis) spectra and steady photoluminescence (PL) spectra of raw $FACsPbI_3$ and $NH_3$-$FACsPbI_3$ perovskite films, respectively.

The scanning electron microscope (SEM) image (Fig. 3b) shows that there are some micrometer-scale voids in the raw $FACsPbI_3$ film. While the $NH_3$-$FACsPbI_3$ film is dense and smooth (Fig. 3c), resulting from the self-leveling of the liquid intermediate state and the quick desorbing process of the $NH_3$. This gas post-healing treatment with the self-leveling behavior is essentially different from the traditional post-treatment using DMF[37], MASCN vapor[25], etc. The corresponding atomic force microscope (AFM) images (Supplementary Fig. 29) show that the root mean square (RMS) roughness of $NH_3$-$FACsPbI_3$ film is only 9 nm over a $20 \times 20$ $\mu m^2$ area, which is even lower than that of the film prepared by the traditional antisolvent method (24 nm) (denoted as anti-$FACsPbI_3$).

X-ray diffraction (XRD) patterns in Fig. 3d and e show that the $NH_3$ post-healing strategy can improve the orientation of the perovskite film, and the diffraction intensity of (110) peak of the $NH_3$-$FACsPbI_3$ perovskite film is about 100 times stronger than that of the raw $FACsPbI_3$ perovskite film. Besides, $NH_3$ post-healing decreases the full-width half-maximum (FWHM) of the (110) peak from 0.203° to 0.114° (Fig. 3e). As shown in ultraviolet-visible (UV-Vis) optical absorption spectra (Fig. 3f) and their Tauc plot curves (Supplementary Fig. 30), the raw and $NH_3$-$FACsPbI_3$ perovskite films present almost the same absorption edge and band gap of 1.54 eV. While the absorbance of $NH_3$-$FACsPbI_3$ perovskite film exhibits an obvious increase compared to that of raw perovskite film, resulting from the removal of voids. Photoluminescence (PL) peak of the $NH_3$-$FACsPbI_3$ film is similar to that of the anti-$FACsPbI_3$ film (Supplementary Fig. 31), but obviously blue shift and narrow in comparison with the raw $FACsPbI_3$ film, which indicates a low trap density of $NH_3$-$FACsPbI_3$ and anti-$FACsPbI_3$ films in comparison with the raw film[38]. The reduced defect density is also evidenced by the charge-limited current (SCLC) and time-resolved photoluminescence (TRPL) results in Supplementary Fig. 32. The calculated defect density from the SCLC curves decreases from $3.97 \times 10^{16}$ to $9.62 \times 10^{15}$ $cm^{-3}$. And the TRPL results show that the $NH_3$-$FACsPbI_3$ film has a much longer trap-assisted nonradiative lifetime ($\tau = 5.6$ $\mu s$) than the Raw $FACsPbI_3$ film ($\tau = 3.2$ $\mu s$).

**Device performance evaluation**

The planar devices with a configuration of FTO/$TiO_2$-$SnO_2$/perovskite/2,2′,7,7′-tetrakis($N,N$-di-p-methoxyphenyl-amine)−9,9′-spirobifluorene (Spiro-OMeTAD)/Au are fabricated. The champion solar cell (Fig. 4a) based on $NH_3$-$FACsPbI_3$ film displays a PCE of 23.21%, with open-circuit voltage ($V_{OC}$) of 1.16 V, short-circuit current density ($J_{SC}$) of 24.65 mA/$cm^2$, fill factor (FF) of 81.20%, demonstrating obvious improvement in comparison with the PCE of 10.92% of the control device based on raw $FACsPbI_3$ film and comparable performance compared to the devices (PCE = 22.99%) based on the anti-$FACsPbI_3$ film in our lab (Supplementary Fig. 33). The certified PCE of $NH_3$-$FACsPbI_3$ PSC is 22.22% (Supplementary Fig. 34). Meanwhile, there is less hysteresis for the $NH_3$-$FACsPbI_3$ device (Supplementary Fig. 35). The stabilized power output (SPO) of $NH_3$-$FACsPbI_3$ champion cells is 22.93%, shown in Supplementary Fig. 36. The $J_{SC}$ of $NH_3$-$FACsPbI_3$ PSC is comparable with the integrated $J_{SC}$ from EQE results (24.41 mA/$cm^2$) in Fig. 4b. The distribution histogram of PCE based on 50 $NH_3$-$FACsPbI_3$ devices at reverse scan direction (Fig. 4c, Supplementary Table 1) shows excellent reproducibility of high-performance $NH_3$ post-healing devices. The cross-sectional SEM image of $NH_3$-$FACsPbI_3$ PSC shows the uniform thickness and good interfacial contact of perovskite film with transport layers (Supplementary Fig. 37). Combined $NH_3$ gas healing and the doctor-blading technology (Suzhou GCL Nano Co. Ltd.), a prototype PSC module consisting 10 cells in series connection achieves a PCE of 20.61% (Fig. 4d) and the certified PCE of the module is 19.38% (Supplementary Fig. 38) with an estimated active area of 14.00 $cm^2$, which is comparable with the reported highest efficiencies in perovskite modules[39–41].

To further evaluate the device stability based on $NH_3$ post-healing treatment, the high-efficiency $NH_3$-$FACsPbI_3$, anti-$FACsPbI_3$, anti-$FA_{0.90}Cs_{0.05}MA_{0.05}PbI_3$ (anti-FMCsPbI$_3$), and anti-$MAPbI_3$ PSCs are employed. As the low PCE, raw $FACsPbI_3$ device has not been chosen to study its stability. The shelf stability of unencapsulated devices in Fig. 4e shows that $NH_3$-$FACsPbI_3$ device has negligible performance loss after 320 days of storage, which is similar to the

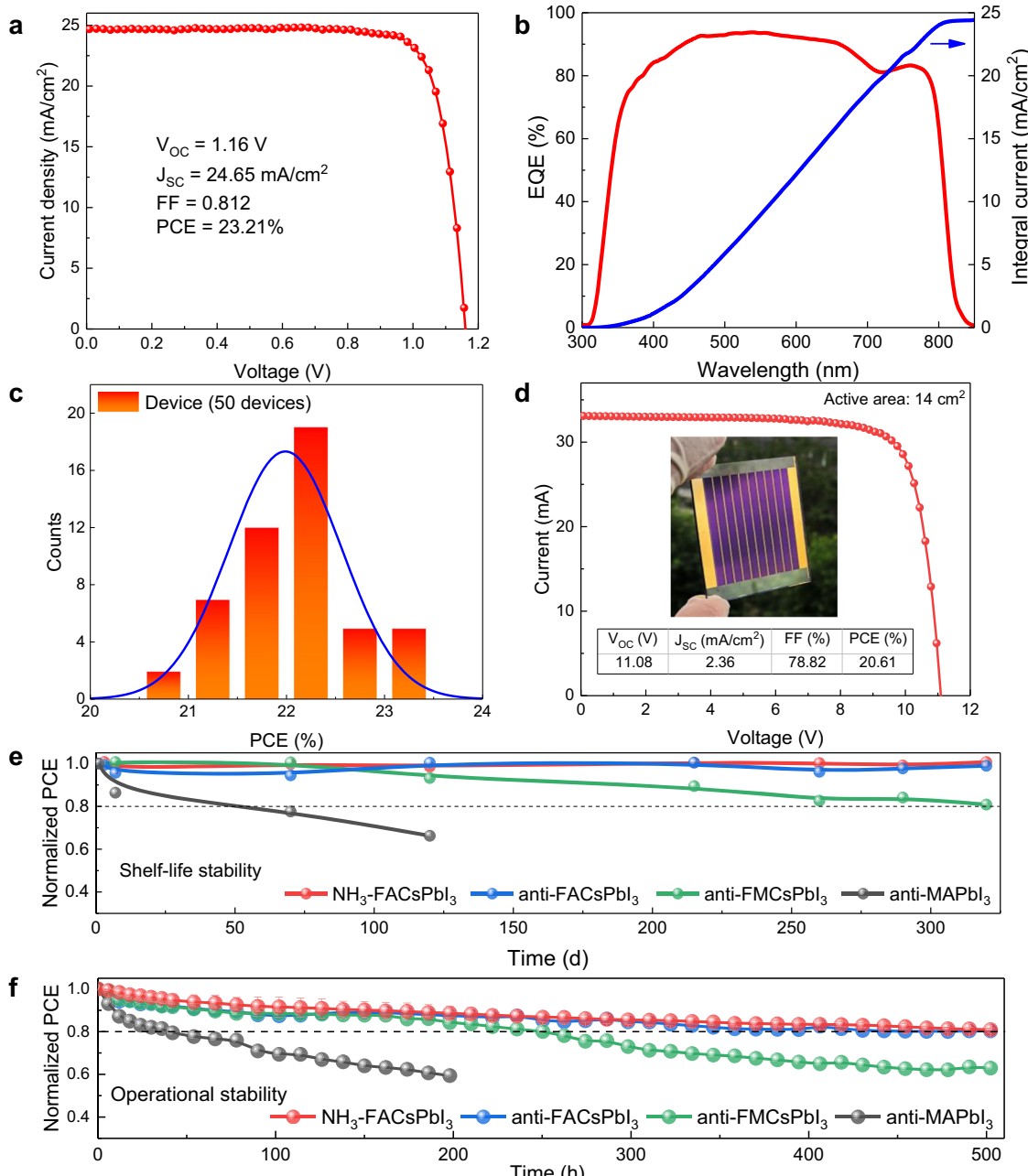

**Fig. 4 | Device performance and stability. a** Current density–voltage (*J-V*) curve of the champion PSCs based on NH$_3$-FACsPbI$_3$. **b** External quantum efficiency (EQE, red) and the integrated short-circuit current density (blue) of the champion NH$_3$-FACsPbI$_3$ device. **c** Histogram of the power conversion efficiency for 50 NH$_3$-FACsPbI$_3$ PSCs. **d** *J-V* curve of NH$_3$-FACsPbI$_3$ PSC module. **e** Normalized power conversion efficiency (PCE) of unencapsulated NH$_3$-FACsPbI$_3$, anti-FACsPbI$_3$, anti-FMCsPbI$_3$, and anti-MAPbI$_3$ devices with the storage time of 320 days under 10–30% relative humidity at room temperature. **f** Evolution of the normalized PCE over time measured by maximum power point tracking of unencapsulated NH$_3$-FACsPbI$_3$, anti-FACsPbI$_3$, anti-FMCsPbI$_3$, and anti-MAPbI$_3$ devices under light soaking with full solar intensity. Standard deviation (error bar) was calculated from three individual devices in the same batch.

anti-FACsPbI$_3$ devices and much better than anti-FMCsPbI$_3$ and anti-MAPbI$_3$. The XRD and UV-Vis spectra of these aged perovskite films in 320-day-aged PSCs (Supplementary Figs. 39–41) show that NH$_3$-FACsPbI$_3$ film keeps its strong light absorption, and has few PbI$_2$. Meanwhile, NH$_3$-FACsPbI$_3$ device still maintains decent cross-sectional morphology with clear grain boundaries (Supplementary Fig. 42). What's more, the maximum power point (MPP) of the unencapsulated NH$_3$-FACsPbI$_3$, anti-FACsPbI$_3$, anti-FMCsPbI$_3$, and anti-MAPbI$_3$ PSCs is tracked under white light emitting diode (LED) irradiation (Fig. 4f, Supplementary Fig. 43) with an intensity equivalent to 1 sun in N$_2$ atmosphere. After 500 h of continuous light soaking, the NH$_3$-FACsPbI$_3$ device maintains 80% of its initial efficiency,

and the stability trend of MPP tracking is consistent with that of shelf stability in these devices.

## Discussion

We have systematically studied the transimination reactions between FA/RA gases and FAI salt, and further developed an NH$_3$ gas post-healing strategy for upscale fabrication of high-quality FA-based perovskite films. The low operating temperature during NH$_3$ post-healing process is vital to enable FA-based perovskite materials to absorb enough NH$_3$ molecules and transform into a flowable inter-mediate state. Based on this strategy, the champion device achieves a PCE of 23.21% (Certified PCE of 22.22%), and the PCE of the perovskite

module is up to 20.61%. Meanwhile, the device stability based on this strategy is also comparable with that of the state-of-the-art anti-solvent method. This $NH_3$ gas post-healing technology is compatible with established commercial technologies to efficiently remove voids in the raw FA-based films and therefore opens a direction for fabrication of large-scale high-efficient FA-based PSCs.

## Methods

### Materials

$N,N$-dimethylformamide (DMF, anhydrous 99.8%), dimethyl sulfoxide (DMSO, anhydrous 99.9%) were procured from Sigma-Aldrich. Formamidine hydrochloride (FACl, 96%), Ethylamine Hydrochloride (EACl, 98%), $n$-Propylamine, (PA, 99%), $n$-Butylamine (BA, 98%), and Lead iodide ($PbI_2$, >98.0%) was purchased from TCI. Formamidinium iodide (FAI, 99.5%), Methanaminium iodine (MAI, 99.5%), Cesium iodide (CsI, 99.99%), and Spiro-OMeTAD (99.8%) were purchased from Xi'an Polymer Light Technology Corp. (PLT). Ammonium chloride ($^{15}NH_4Cl$, 10 atom% $^{15}N$ labelled, ≥98.5%) was purchased from Macklin. Ammonium chloride ($^{15}NH_4Cl$, 99 atom% $^{15}N$ labelled, >98%) and DMSO-$d_6$ (99 atom% D labelled) were purchased from Cambridge Isotope Laboratories, Inc. (CIL). Sodium hydroxide (NaOH, 96%), Methylamine alcohol solution (30–33 wt. % in anhydrous ethanol), and Ammonium Hydroxide ($NH_3 \cdot H_2O$, 25~28%) were purchased from Sinopharm Chemical Reagent Co., Ltd. The above chemicals were used as received without any purification.

### Material synthesis

For the synthesis of N-methyl formamidinium iodide (MFAI), Methylamine alcohol solution (30–33 wt. % in anhydrous ethanol, 425 μl, about 2.8 mmol) was added dropwise to a solution of formamidinium iodide (FAI, 516 mg, 3 mmol) in 30 ml anhydrous ethanol under nitrogen protection, and the mixture was stirred at 0 °C for 2 h. Subsequently, the solvent was evaporated under reduced pressure and the residue was recrystallized in the mix solvent of isopropanol and ethyl acetate in glove box to obtain the MFAI powder. The synthesis of N, N'-dimethyl formamidinium iodide (DMFAI) was similar to that of MFAI, but the amount of the methylamine alcohol solution was increased to 1 ml.

### NMR sample fabrication

For solution $^1H$ NMR characterization of the sample in Table 1 (No.1), the mixture of 5.0 g FACl and 5.0 g NaOH was reacted at RT in a conical flask, and the generated gas was collected by 0.5 ml DMSO-$d_6$. For samples in Table 1 (No. 2–4), the mixture of 5.0 g FACl and 5.0 g NaOH was reacted at 150 °C, and the generated gas was collected by 0.5 ml DMSO-$d_6$, 1 ml HOAc or 0.2 g FAI powder at RT. Samples No. 3 and 4 were vacuumed before dissolving in DMSO-$d_6$. For samples in Table 1 (No. 5–12), the gas sources of $R$-$NH_2$ were heated at 60 °C, and the generated gases were collected by 0.2 g FAI powder. The gas sources of $R$-$NH_2$ were methylamine alcohol solution, the mixture of EACl and NaOH, PA, BA, $NH_3 \cdot H_2O$, MA aqueous solution, EA aqueous solution, or $H_2O$, respectively. The samples (No. 5–12) were kept at 60 °C for 6 h, and then vacuum treated to remove the low boiling point components. For the sample in Table 1 (No. 13), the $NH_3$ gas was collected with 0.2 g FAI powder. The sample was kept at RT for 6 h and then vacuumed. The obtained solid powders in Table 1 (No. 3–13) were dissolved in DMSO-$d_6$ for $^1H$ NMR characterization.

For solution $^{15}N$ NMR characterization, $^{15}NH_3$ was produced from the reaction between 5.0 g $^{15}NH_4Cl$ (10% $^{15}N$ labelled) and 5.0 g NaOH at 60 °C, and the generated gas was collected by 0.5 ml DMSO-$d_6$. FAI ($^{15}N$) was produced based on the reaction between 0.2 g FAI and excess $^{15}NH_3$ at RT for 6 h, followed by vacuum treatment. FAI·x$^{15}NH_3$, $PbI_2$·x$^{15}NH_3$, and FAPbI$_3$·x$^{15}NH_3$ were generated between the reaction between 0.2 g FAI, 0.2 g $PbI_2$, or 0.2 g FAPbI$_3$ precursor powders and excess $^{15}NH_3$ at RT for 6 h, but without vacuum treatment.

The above samples were directly dissolved in DMSO-$d_6$ for $^{15}N$ NMR measurement.

The NMR sample fabrication of FAPbI$_3$·x$^{15}NH_3$ for the solid-state NMR spectrum was similar to that of the solution $^{15}N$ NMR spectrum, but $^{15}N$ isotope source for preparing $^{15}NH_3$ was used $^{15}NH_4Cl$ (99% $^{15}N$ labelled) as the requirement of high $^{15}N$ concentration for solid-state NMR.

### NMR measurements

The solution $^1H$ NMR spectra were measured with a Bruker AVANCE III 600 instrument operating at 600.13 MHz at 298 K. Chemical shifts d (ppm) were referenced to the internal solvent signals.

The solution $^1H$-decoupled $^{15}N$ NMR spectra (pules program zgig) were acquired using a Bruker Avance III 600 instrument operating at 600.13 MHz for $^1H$ and 60.82 MHz for $^{15}N$. The $^{15}N$ chemical shifts were determined from 1 M urea in DMSO (−303.20 ppm, 10% $^{15}N$ labelled) as external standard reference [42]. The relaxation delay was 10 s, the acquisition time was 1.3 s, and 128 scans were accumulated.

The solid-state NMR experiment was performed on a Bruker AVNCE 400 spectrometer operating at 399.95 MHz for $^1H$ and 40.54 MHz for $^{15}N$. The obtained sample was packed into a 7 mm $ZrO_2$ rotor. The $^{15}N$ chemical shifts were determined from the $^{15}NH_4Cl$ (−341.17 ppm, 10% $^{15}N$ labelled)[43]. The HPEDC pulse sequence was used. The acquisition time was 50 ms, the spin rate was 3 kHz, and the recycle delay was 5 s. The two-pulse phase-modulation (TPPM) decoupling was used during the acquisition.

### Perovskite film fabrication

Raw FACsPbI$_3$ films: The perovskite precursor was prepared by mixing $PbI_2$: FAI: CsI (1:0.9:0.1 molar ratio) in DMF solvent (1.40 M). Raw FAPbI$_3$ films: The perovskite precursor was prepared by mixing $PbI_2$: FAI (1:1 molar ratio) in DMF solvent (1.40 M). The raw FACsPbI$_3$ and FAPbI$_3$ films were simply fabricated by one-step spin-coating at 4000 rpm for 30 s and then heated at 140 °C for 20 min.

$NH_3$-FACsPbI$_3$ film: After spin-coating, the raw FACsPbI$_3$ film was annealed at 140 °C for 5 min, and then transferred into a home-made chamber with a temperature from 20 to −15 °C. $N_2$ gas was used to remove the moisture-laden air in the chamber. Subsequently, $NH_3$ gas was quickly introduced into the chamber and maintained for ~5 s. Then the $NH_3$ gas is removed from the chamber and the film was further annealed at 140 °C for 20 min. For $NH_3$-FAPbI$_3$ film, the film treatment process is the same as above, and the temperature of the homemade chamber is −15 °C for $NH_3$ post-healing process.

For anti-FACsPbI$_3$, anti-FMCsPbI$_3$, and anti-MAPbI$_3$ films, the perovskite precursors were prepared in DMF: DMSO (3:7, 1.40 M) by stoichiometric ratios. The perovskite solution was spin-coated in a two-step at 1000 rpm and 4000 rpm for 10 s and 30 s, respectively. During the second step, 300 μl anisole was drop-casted quickly at the tenth second of the second step. The anti-MAPbI$_3$ perovskite film was then heated at 100 °C for 20 min, and the other films were heated at 140 °C for 20 min.

The α-FAPbI$_3$, δ-FAPbI$_3$, MFAPbI$_3$, and DMFAPbI$_3$ films were fabricated from their DMF solution with a concentration of 1.40 M using chlorobenzene as the antisolvent. The α-FAPbI$_3$, MFAPbI$_3$, and DMFAPbI$_3$ films were obtained after heating treatment at 140 °C for 20 min. The δ-FAPbI$_3$ film was achieved by exposing α-FAPbI$_3$ film in the air (RH > 50%) until fully transformed into yellow. The MA-FAPbI$_3$ film was prepared by the MA$^0$ gas post-healing the α-FAPbI$_3$ film and then heated at 140 °C for 20 min.

The annealing process of all thin films is in ambient air conditions (30–40% humidity).

### Device fabrication

The methods described are similar to that reference 44. Fluorine doped tin oxide (FTO)-coated glass (2.20 mm, 7Ω sq$^{-1}$) was used as the

substrate for the devices. The compact $TiO_2$ layer (~10 nm) was deposited by atomic layer deposition (ALD) for 200 cycles and annealed at 500 °C for 30 min in ambient air. For ALD $TiO_2$ deposition, titanium (IV) isopropoxide (TTIP) and $H_2O$ as Ti and O sources, respectively. The TTIP precursor was held at 75 °C. Pulse/exposure/purge times of 1 s/8 s/25 s were used for the TTIP and 0.1 s/8 s/25 s for $H_2O$ precursor, and the deposition temperature was set to 120 °C. On top of the c-$TiO_2$ layer, $SnO_x$-Cl layer was deposited by spin-coating at a speed of 3000 rpm for 30 s from an aged $SnCl_4$ aqueous solution (1:75 with deionized water by volume), followed by a sintering heat-treatment of 200 °C for 30 min in air and then transferred to the glove box for device fabrication. The perovskite layers were fabricated as described above. For the surface passivation, 1 mg ml$^{-1}$ of Phenyltrimethylammonium tribromide (PTAB) solution in isopropanol was spin-coated on these perovskite films at 4000 rpm for 30 s.

The spiro-OMeTAD chlorobenzene solution (72.3 mg ml$^{-1}$) with 28.8 μl 4-tert-butylpyridine (96%, Aldrich-Sigma) and 17.5 μl lithium bis(trifluoro-methanesulfonyl) imide (Li-TSFI, Aldrich-Sigma) solution (520 mg Li-TSFI (98%) in 1 ml acetonitrile (99.8%, Aldrich-Sigma)) was spin-coated on top of the perovskite film at 3000 rpm for 30 s. The devices were put into a dry-air box (RH < 3%) for 12 h. Finally, 80 nm thick Au electrode was thermally evaporated.

For fabricating modules, P1 etching process was pre-patterned on FTO glass (5 cm × 5 cm) with a 1064 nm fiber laser (Han's laser). The laser power ratio, laser duty cycle, and laser frequency were 30%, 5%, and 50 kHz, respectively. Then, patterned FTO substrates were cleaned and treated by UV Ozone Cleaner (Ossila) for 15 min. The $TiO_2$/$SnO_2$, PTAB, and Spiro-OMeTAD layers were prepared with the same procedure as presented above. The large-size raw $FACsPbI_3$ perovskite film was prepared through the doctor-blading method (provided by Suzhou GCL Nano Co. Ltd.). The $NH_3$ post-healing strategy was carried out on it to form a large-scale $NH_3$-$FACsPbI_3$ film. For P2 etching process, the laser used was a 532 nm laser with a laser power ratio of 65%, a laser duty cycle of 5%, and a laser frequency of 100 kHz. 80 nm thick Au electrodes were thermally evaporated under vacuum to complete the modules fabrication. Finally, P3 etching used the same laser with P2 with a laser power ratio of 50%, a laser duty cycle of 5%, and a laser frequency of 100 kHz. P4 is an etching procedure for cleaning the edge of the modules, the laser used in P4 is the same with P1 with a laser power ratio of 40%, the laser duty cycle of 10%, and a laser frequency of 100 kHz.

### Perovskite film and device characterization

The characterization described are similar to that reference 44. XRD spectra were measured by Ultima IV of Rigaku with Cu Kα radiation (1.5406 Å). The UV-Vis absorbance spectra were measured by QE Pro (Ocean Optics). Top view, cross-section SEM images were obtained with a field-emission SEM (S-4800, Hitachi). Steady PL spectra were recorded on QE Pro excited at 460 nm. AFM measurements were performed in contact mode (5400, Agilent). EQE measurement was calculated using certified incident photon to current conversion efficiency equipment from Enlitech (QE-R). Time-resolved photoluminescence (TRPL) experiments were performed by Steady State and Transient State Fluorescence Spectrometer (Edinburgh FLS980). The testing conditions as the films were photoexcited at 483.6 nm pulse width ~ 118.6 ps, 5 mW/pulse, and emission were collected on the surface side of the film (perovskite/glass substrate).

*J-V* curves of the as-fabricated PSCs were measured using a SourceMeter (Keithley 2400) under simulated one-sun AM 1.5 G 100 mW cm$^{-2}$ intensity (Oriel Sol3A Class AAA, Newport) with a scan rate of 200 mV/s (the voltage step is 20 mV with no delay time) from reverse and forward two scanning directions in air condition around 25 °C. The typical active area of PSCs is 0.09 cm$^2$ defined by a metal mask. The area of the mask (0.08713 cm$^2$) used for certification was certified by the National Institute of Metrology, China, No. CDjc2021-10891. The intensity of one-sun AM 1.5 G illumination was calibrated using a Si-reference cell certified by the National Renewable Energy Laboratory. SPO is measured by tracking the current under a fixed voltage which is decided by the voltage of maximum power point at the *J-V* curve. Here, the fixed voltage of $NH_3$-$FACsPbI_3$ device is 1.002 V.

For the stability tests, all PSCs were without encapsulation. For shelf-life stability, the devices were stored in a dark environment with a humidity of 10–30%, and the photovoltaic performance of PSCs was measured every ten days. The operational stability was performed using a stability setup (LC Auto-Test 24, Shenzhen Lancheng Technology Co., Ltd.), tested under continuous light illumination and maximum power point tracking (controlled and monitored to be 15 °C). The light source consisted of an array of white LEDs powered by a constant current. The LED type is MG-A200A-AE with an emission spectrum of 400–750 nm (Supplementary Fig. 40). Equivalent sun intensities were calibrated using a calibrated Si-reference cell. During aging, the device is connected with a 100 Ohm load resistance. The PSCs were masked and placed inside a sample holder purged with continuous $N_2$ flow. *J-V* curves with reverse voltage scans were recorded every 12 h during the whole operational test.

### Reporting summary

Further information on research design is available in the Nature Research Reporting Summary linked to this article.

### Data availability

Data that support the findings of this study are available in Supplementary Data Files in the Supplementary Information section. Source data are provided with this paper.

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

## Acknowledgements

S.P.P. acknowledges funding from the Youth Innovation Promotion Association CAS (Y201944), Funding of Young Taishan Scholars (tsqn201812110), Natural Science Foundation of Shandong Province (ZR2020KB001), and funding from Shandong Energy Institute (SEI I202129), X.W. thanks the Natural Science Foundation of Shandong Province (ZR2020KB001). Z.P.S. thanks the National Natural Science Foundation of China (51902324). We thank Suzhou GCL Nano Co. Ltd. for the fabrication of large-sized perovskite films and the Qingdao Key Lab of Solar Energy Utilization & Energy Storage Technology.

## Author contributions

L.W., G.L.C., and S.P.P. conceived and planned the experiments with additional input from Z.W.W. and Z.P.S. Z.P.L. fabricated all samples and devices, and performed and analyzed the SEM, XRD experiments, the device efficiency, and stability experiments. X.W. performed and analyzed the NMR experiments. L.Z.H., Y.R., C.C., and D.C.L. carried out the gas absorbing and desorbing test. Q.Q.Z. and X.H.S. assisted with film fabrication and optical measurement. C.Y.G. and B.Q.Z. helped to perform and analyze NMR measurements. X.Z.W. helped to perform TRPL measurements. Z.P.L., Z.W.W., and S.P.P. took the lead in drafting the manuscript and compiled the figures. All authors discussed the results and provided feedback on the manuscript.

## Competing interests

The authors declare no competing interests.
