## [Peer Review File · Nature Communications]

Ammonia for post healing of formamidinium-based perovskite filmsEditorial Notes : This manuscript has been previously reviewed at another journal that is not operating a transparent peer review scheme. This document only contains reviewer comments and rebuttal letters for versions considered at *Nature Communications*. Parts of this Peer Review File have been redacted as indicated to remove third-party material where no permission to publish could be obtained.

REVIEWER COMMENTS

Reviewer #2 (Remarks to the Author):

This is an interesting study which could be useful for the fabrication of large-area FA-based PSCs. The following questions need to be properly addressed before publication.

1. The authors consider a lower trap density of perovskite films after the treatment simply based on the blue shifted and narrowed PL peaks. However, it would be a bit risky to draw to such a conclusion by just PL data because differences in stoichiometry, thickness as well as other factors would also show an impact on the shape and position of PL spectra. I would recommend to further consolidate the claim of decreased trap density by doing other characterizations, such as pump-probe terahertz or TRMC.

2. How does the NH₃ post healing process affect the thickness of the films considering there have a liquid intermediate during the process.

3. I appreciate that the statistics have now been included in the stability data. However, details, such as the number of devices and batches, the definition of the error bars, should be properly described in the figure caption as well.

Reviewer #4 (Remarks to the Author):

The review is attached.

The article Li et al. reports a strategy for fabricating perovskite films using ammonia treatment to induce high uniformity. The authors use NMR to elucidate the mechanism of action of NH_3 and other amines, and fabricate high efficiency and highly stable devices, comparable to those fabricated using conventional antisolvent approaches.

While the experimental results are of high quality and the authors provide very important insights about the molecular level mechanism of action of the proposed strategy, the reviewer feels that the level of novelty does not warrant publication in Nature Energy, since similar gas healing approaches have previously been proposed, for example, based on methylamine (ref. 17). The work will be of high interest to the broad chemistry readership in another journal, such as Nat. Commun.

Before it can be considered for publication, the authors should substantially improve the discussion surrounding the NMR data, since many of the statements are currently given with no explanation or reference, as detailed below:

“We think that the high boiling point of FA^0 likely makes FA^0 molecules difficult to diffuse into DMSO-d_6 solvent.” – why would boiling point affect solubility of FA^0 ? As a highly polar molecule, formamidine should be well soluble in DMSO. There is no evidence in the data that FA^0 does form. This makes the caption of Table 1 “The synthesis of formamidine (FA^0) and chemical reactions between FA^0 /amines/ NH_3 gases and FAI salt” and the surrounding discussion unsubstantiated.

There are also a number of issues in the interpretation of the NMR data: the assignment is insufficient, some peaks are not discussed and references are not cited to corroborate the proposed assignments. Please assign each signal to the moiety it corresponds to on a given molecule (rather than simply marking the peaks with asterisks, as one would do for an XRD pattern). As is, the assignments are arbitrary and cannot be treated as valid. Please also label the peak of the residual solvent protons.

Extended Data Figure 3: 0.70 ppm is suggested to be NH_3 while naively a much larger chemical shift is expected due to the high electronegativity of nitrogen. <https://pubs.acs.org/doi/abs/10.1021/acs.jpcc.7b09805> indeed shows that NH_4^+ resonates around 7 ppm). Please discuss this discrepancy. What is the signal at 3.48 ppm? Increasing the pH, as suggested in the work above, would substantially facilitate interpretation of the spectra in general by narrowing the peaks of exchangeable (NH, OH) protons and evidencing their J-coupling to the neighbouring protons.

Extended Data Figure 4: FA has either two or three ^1H peaks depending on pH (CH and NH at 8 ppm and between 8-9 ppm, respectively, see Figure 5 in the work referenced above). The data shown here do not support the presence of FA, as currently assigned. NH_3 in this case is assigned to a peak at 3.54 ppm, in contradiction to Extended Data Figure 3. S-triazine and formamide are assigned with no explanation or reference given. Finally, there are a number of peaks in the 7.5-9 ppm region which are not assigned at all.

Extended Data Figure 5: Please assign each signal to the corresponding moiety. The peak at 3.2 ppm in panel 'a' is not assigned.

Extended Data Figure 6: Please assign each signal to the corresponding moiety. The peak at 3.42 ppm is not assigned.

Extended Data Figure 7: Please assign each signal to the corresponding moiety. There are a large number of peaks which are not assigned.

Extended Data Figure 8: Please assign each signal to the corresponding moiety and explain the observed J-couplings in the context of the assignment.

Extended Data Figure 9-13, 16, 19-23 : As above.

Extended Data Figure 24: what is this assignment based on? Comparing with the liquid-state spectra in Figure 1, the ^{15}N of FAI resonates at higher frequencies than that of $\text{PbI}_2 \cdot x\text{NH}_3$. Why is the assignment in the solid state reversed?

Extended Data Figure 19-23: please assign the reference (urea) signal.

Figure 1: regarding "FAI treated with $^{15}\text{NH}_3$ and vacuumed ($\text{FAI}(^{15}\text{N})$)" – the authors attribute the ^{15}N peak to ^{15}N -labelled FAI. According to Scheme 1, the reaction of FAI with NH_3 should yield NH_4I and FA^0 , which is then evacuated. Table 1 and Ext. Data Fig. 16 show that the reaction product is FAI, but this creates confusion (since Scheme 1 states that R can be equal H). Based on Scheme 1 it is also unclear how the ^{15}N -labelled nitrogen of NH_3 would end up in FAI. Figure 1c clarifies this issue but it is still in apparent contradiction to the generality of Scheme 1.

Figure 1b also contains a number of peaks which are not assigned (two small peaks in the $\text{FAI} \cdot x\text{NH}_3$ spectrum), suggesting that the reaction is more complex. Why is there no signal from the $^{15}\text{NH}_3$ species in this case while $\text{FAPbI}_3 \cdot x\text{NH}_3$ contains it?

"The hydrolysis reactions of FA^0 , RFA^0 and DRFA^0 lead to the formation of volatile HCOOH and RA^0 " – what is this conclusion based on? How was the presence of the volatile products concluded?

"The s-triazine with a boiling point of $114\text{ }^\circ\text{C}$ is attributed to the transimination reaction of three FA^0 molecules along with the formation of NH_3 " – this is not a transimination reaction but condensation: please refer to <https://pubs.acs.org/doi/abs/10.1021/acs.jpcc.7b09805> and the references within (in particular by Schaeffer et al., ref. 20).

Finally, since NMR is not a commonly used technique in the field of halide perovskite fabrication, it would be beneficial to include a brief introduction to its main tenets and differences relative to XRD etc. A few relevant recent reviews on the subject include

<https://www.nature.com/articles/s41570-021-00309-x>

<https://pubs.acs.org/doi/abs/10.1021/jacs.0c07338>

The added data and corresponding discussions are emphasized in the revised manuscript

Referees' comments:

Reviewer #2 (Remarks to the Author):

This is an interesting study which could be useful for the fabrication of large-area FA-based PSCs. The following questions need to be properly addressed before publication.

1. The authors consider a lower trap density of perovskite films after the treatment simply based on the blue shifted and narrowed PL peaks. However, it would be a bit risky to draw to such a conclusion by just PL data because differences in stoichiometry, thickness as well as other factors would also show an impact on the shape and position of PL spectra. I would recommend to further consolidate the claim of decreased trap density by doing other characterizations, such as pump-probe terahertz or TRMC.

Response:

We thank the reviewer for all the helpful comments and suggestions.

In the revised manuscript, we have added the charge-limited current (SCLC) and time-resolved photoluminescence (TRPL) measurements to prove the reduction of the trap density of the perovskite films after the NH₃ treatment.

“The reduced defect density is also evidence by the charge-limited current (SCLC) and time-resolved photoluminescence (TRPL) results in Supplementary Fig. 30. The calculated defect density from the SCLC curves decreases from 3.97×10^{16} to 9.62×10^{15} cm⁻³. And the TRPL results shows that the NH₃-FACsPbI₃ film has a much longer PL lifetime (132.78 ns) than the Raw FACsPbI₃ film (10.06 ns).” The above conclusion has been added in the revised manuscript.

Supplementary Figure 30 | (a, b) J-V characteristics derived from the SCLC measurements with a structure of FTO/SnO₂/perovskite/PCBM/Au and (c) Time-resolved photoluminescence decay curves of the Raw and NH₃-FACsPbI₃ films.

2. How does the NH_3 post healing process affect the thickness of the films considering there have a liquid intermediate during the process.

Response:

The NH_3 gas post healing treatment only refers to the elimination of voids and the reduction of the roughness of the film. The formation of liquid intermediate phase in the NH_3 gas post healing treatment leads to the structural collapse of raw perovskite film and then rearrangement of the high uniform film as the evaporation of NH_3 . The morphology evolution after the NH_3 post healing is clearly presented in Supplementary Fig. 35.

Supplementary Figure 35 | Cross-sectional SEM images of (a) Raw FACsPbI₃ and (b) NH₃-FACsPbI₃ PSCs.

3.I appreciate that the statistics have now been included in the stability data. However, details, such as the number of devices and batches, the definition of the error bars, should be properly described in the figure caption as well.

Response:

Thank you for the suggestion. The figure caption has been reorganized including all of the contents mentioned above.

Referee #4 (Remarks to the Author):

The article Li et al. reports a strategy for fabricating perovskite films using ammonia treatment to induce high uniformity. The authors use NMR to elucidate the mechanism of action of NH_3 and other amines, and fabricate high efficiency and highly stable devices, comparable to those fabricated using conventional antisolvent approaches.

While the experimental results are of high quality and the authors provide very important insights about the molecular level mechanism of action of the proposed strategy, the reviewer feels that the level of novelty does not warrant publication in Nature Energy, since similar gas healing approaches have previously been proposed, for example, based on methylamine (ref. 17). The work will be of high interest to the broad chemistry readership in another journal, such as Nat. Commun.

Response:

We thank the reviewer for the positive comment on our insight about the molecular level mechanism. There may be a misunderstanding. Our paper was submitted to Nat. Commun., not Nature Energy.

Before it can be considered for publication, the authors should substantially improve the discussion surrounding the NMR data, since many of the statements are currently given with no explanation or reference, as detailed below:

“We think that the high boiling point of FA^0 likely makes FA^0 molecules difficult to diffuse into DMSO-d_6 solvent.” – why would boiling point affect solubility of FA^0 ? As a highly polar molecule, formamidine should be well soluble in DMSO. There is no evidence in the data that FA^0 does form. This makes the caption of Table 1 “The synthesis of formamidine (FA^0) and chemical reactions between FA^0 /amines/ NH_3 gases and FAI salt” and the surrounding discussion unsubstantiated.

Response:

“We think that the high boiling point of FA^0 likely makes FA^0 molecules difficult to diffuse into DMSO-d_6 solvent.” In this context, we performed the experiment at room temperature (Supplementary Fig. 1a), thus, the high boiling point of FA^0 cannot volatilize and diffuse into the DMSO-d_6 solvent as gas. The nonvolatile FA^0 undergoes a hydrolysis reaction in the reactor to produce ammonia (NH_3). Thus, only NH_3 signal was detected in the ^1H NMR (Table 1(No.1) and Supplementary Fig. 2). When temperature of the reactor and the gas pipeline is increased to $150\text{ }^\circ\text{C}$, FA^0 can volatilize and enter the DMSO-d_6 solvent. At the same time, some volatile by-products also enter the DMSO-d_6 solvent, and the main compounds are detected: s-triazine, formamide and FA^0 (Table1 (No.2) and Supplementary Fig. 3). The problem here is whether the FA^0 can volatilize easily and diffuse into the DMSO-d_6 solvent, which does not involve the relationship between boiling point and solubility of FA^0 .

Supplementary Figure 1 | (a) Schematic illustration of the chemical reaction between FACl and NaOH powders with the gas collection by DMSO-d₆ under RT or 150 °C

There are also a number of issues in the interpretation of the NMR data: the assignment is insufficient, some peaks are not discussed and references are not cited to corroborate the proposed assignments. Please assign each signal to the moiety it corresponds to on a given molecule (rather than simply marking the peaks with asterisks, as one would do for an XRD pattern). As is, the assignments are arbitrary and cannot be treated as valid. Please also label the peak of the residual solvent protons.

Response:

Thank you for the helpful suggestion. We have carefully assigned each signal to the corresponding moiety and relevant references have been cited to corroborate the proposed assignments.

Extended Data Figure 3: 0.70 ppm is suggested to be NH₃ while naively a much larger chemical shift is expected due to the high electronegativity of nitrogen. <https://pubs.acs.org/doi/abs/10.1021/acs.jpcc.7b09805> indeed shows that NH₄⁺ resonates around 7 ppm). Please discuss this discrepancy. What is the signal at 3.48 ppm? Increasing the pH, as suggested in the work above, would substantially facilitate interpretation of the spectra in general by narrowing the peaks of exchangeable (NH, OH) protons and evidencing their J- coupling to the neighbouring protons.

Response:

Thank you for the kind suggestion. To prove the generating of NH₃ in the experiment, the NMR spectrum of DMSO-d₆ directly absorbing NH₃ was performed (Supplementary Fig. 2b) and the result has the almost same signals with Supplementary Fig. 2a. The discrepancy in chemical shift values of NH₃ and NH₄⁺ signals is caused by the difference in electron cloud density on N atoms. The electron cloud density on the N atom in the protonated NH₃ (NH₄⁺) will decrease, and as the result, the chemical shift value will move to low field.

Supplementary Figure 2 | (a) The ¹H NMR spectrum of the reaction product between FACl and NaOH powders under room temperature directly absorbed by DMSO-d₆ (b). Commercial NH₃ gas directly absorbed by DMSO-d₆ under room temperature.

Extended Data Figure 4: FA has either two or three ¹H peaks depending on pH (CH and NH at 8 ppm and between 8-9 ppm, respectively, see Figure 5 in the work referenced above). The data shown here do not support the presence of FA, as currently assigned. NH₃ in this case is assigned to a peak at 3.54 ppm, in contradiction to Extended Data Figure 3. S-triazine and formamide are assigned with no explanation or reference given. Finally, there are a number of peaks in the 7.5-9 ppm region which are not assigned at all.

Response:

Thank you for your suggestion. Due to the multiple reacting possibility of FA⁰ in solution, the products are very complex (*Chemical Reviews*, 1944, 35, 351-425, doi: 10.1021/cr60112a002). For Supplementary Fig. 3, we only assigned the main signals. In the 7.5-9 ppm region, although there are many signals with low peak intensity, which caused by a variety of unstable reaction intermediates, and these signals cannot be completely accurately assigned. We believe that such interpretation of the spectrum does not affect our

explanation of the main reaction process in this system. The signals of s-triazine and formamide are referenced to the SDBS database (https://sdb.sdb.aist.go.jp/sdb/cgi-bin/direct_frame_top.cgi, the print screen as follow). Due to the different deuterium reagents and NMR equipment condition, there are subtle differences in chemical shift values between the reference and our spectra. ^1H NMR characterization of commercialized s-triazine and formamide was also carried out under the same test conditions (Figure R1), and our results were matched. The signal at 7.40 ppm was assigned as hydrogen in N-CH=N group of FA⁰. This is because in FA⁰ which generates after the deprotonation of FAI, the electron cloud density on hydrogen atom in N-CH=N will increase, and the chemical shift value will move to the high field compared with FAI. Furthermore, the HOAc/DMSO solution was used to collect the gas produced in the reaction between FACl and NaOH powders and the NMR spectrum shows the FAAc signal (Supplementary Fig. 4), which also supports our interpretation of Supplementary Fig. 3. Moreover, there was an error in the assignment of the signal at 3.54 ppm in Supplementary Fig. 3, and it should be the signal of the HDO. We have revised it in the text and SI. The strong HDO signal indicates there was some water generating from the neutralization reaction. The activity hydrogen with low pKa value (such as N-H in amidines and ammonium) could deuterium exchange with HDO causing its signals unobserved. In addition, the signals of FA in the reference (<https://pubs.acs.org/doi/abs/10.1021/acs.jpcc.7b09805>) is the NMR data of FAI, which is difference with FA⁰.

[REDACTED]

Figure R1 The print screen (a, b) and ^1H NMR spectra (c, d) of s-triazine and formamide

Extended Data Figure 5: Please assign each signal to the corresponding moiety. The peak at 3.2 ppm in panel 'a' is not assigned.

Response:

Thank you for the suggestion. We have assigned each signal to the corresponding moiety.

"The peaks were assigned as: FAAC, δ 9.94 for (a) and 9.92 for (b)(br, N-H), 7.76(s, N-HC=N), 1.66(s, -CH₃); HDO, δ 3.36 for (a)(br, HDO); DMSO-d₆ residual peak, δ 2.50. The peak at 3.36 ppm in panel 'a' is assigned to HDO."

Extended Data Figure 6: Please assign each signal to the corresponding moiety. The peak at 3.42 ppm is not assigned.

Response:

Thank you for the suggestion. We have assigned each signal to the corresponding moiety.

"The peaks were assigned as: FAI, δ 8.72(br, N-H), 7.85(s, N-HC=N); formamide, δ 7.95(dd, $J_1 = 1.4$ Hz, $J_2 = 13.5$ Hz, H-C=O), 7.43(br, -N-H), 7.15(br, -N-H); HDO, δ 3.54 (br, HDO). DMSO-d₆ residual peak, δ 2.50. The peak at 3.42 ppm is assigned to HDO."

Extended Data Figure 7: Please assign each signal to the corresponding moiety. There are a large number of peaks which are not assigned.

Response:

Thank you for the suggestion. We have assigned each signal to the corresponding moiety.

"The peaks were assigned as: DMFAI, δ 8.95(br, N-H), 7.99(s, N-HC=N), 3.03(s, C=N-CH₃), 2.79(s, C-N-CH₃); MFA, δ 8.00(br, N-H), 2.58(dd, $J_1 = 0.54$ Hz, $J_2 = 4.8$ Hz, N-CH₃); MAI, δ 2.37(s, N-CH₃); HDO, δ 3.32 (br, HDO); DMSO-d₆ residual peak, δ 2.50."

Extended Data Figure 8: Please assign each signal to the corresponding moiety and explain the observed J-couplings in the context of the assignment.

Response:

Thank you for the suggestion. We have assigned each signal to the corresponding moiety.

"The peaks were assigned as: DEFAL, δ 9.34(br, N-H), 8.99(br, N-H), 8.02(d, $J = 6.72$ Hz, N-HC=N), 3.35(q, $J = 7.06$ Hz, C=N-CH₂), 3.18(q, $J = 6.40$ Hz, C-N-CH₂), 1.06(m, N-CH₂-CH₃); DEFA, δ 8.86(br, N-H), 8.24(s, N-HC=N), 3.26(q, $J = 7.20$ Hz, C-N-CH₂), 1.06(m, N-CH₂-CH₃); EAI, δ 7.57(br, N-H), 2.78(q, $J = 7.24$ Hz, C-N-CH₂), 1.06 (m, N-CH₂-CH₃); DMSO-d₆ residual peak, δ 2.50."

Extended Data Figure 9-13, 16, 19-23: As above.

Response:

Thank you for the suggestion. We have assigned each signal to the corresponding moiety.

"Supplementary Figure 8: The peaks were assigned as: DPFAI, δ 8.78(br, N-H), 8.02(s, N-HC=N), 3.30(t, $J = 6.96$ Hz, C=N-CH₂), 3.16(t, $J = 7.05$ Hz, C-N-CH₂), 1.50(m, N-CH₂-CH₂), 0.85(m, N-CH₂-CH₂-CH₃); DPFA, δ 8.25(br, N-H), 8.20(s, N-HC=N), 3.22(m, C-N-CH₂), 1.50(m, N-CH₂-CH₂), 0.85(m, N-CH₂-CH₂-CH₃); PAI, δ 8.78(br, N-H), 2.73(t, $J = 7.50$ Hz, C-N-CH₂),

1.50(m, N-CH₂-CH₂), 0.85(m, N-CH₂-CH₂-CH₃); DMSO-d₆ residual peak, δ 2.50.

Supplementary Figure 9: The peaks were assigned as: DBFAI, δ 9.42(br, N-H), 9.02(br, N-H), 8.06(s, N-HC=N), 3.35(t, J = 7.11 Hz, C-N-CH₂), 3.18(t, J = 7.14 Hz, C-N-CH₂), 1.47(m, N-CH₂-CH₂), 1.25(m, N-CH₂-CH₂-CH₂), 0.82(m, N-CH₂-CH₂-CH₂-CH₃); DBFA, δ 9.02(br, N-H), 8.30(s, N-HC=N), 3.26(t, J = 7.05 Hz, C-N-CH₂), 1.25(m, N-CH₂-CH₂-CH₂), 0.82(m, N-CH₂-CH₂-CH₂-CH₃); BAI, δ 2.76(t, J = 7.59 Hz, C-N-CH₂), 1.25(m, N-CH₂-CH₂-CH₂), 0.82(m, N-CH₂-CH₂-CH₂-CH₃); DMSO-d₆ residual peak, δ 2.50.

Supplementary Figure 10: The peaks were assigned as: NH₄I, δ 7.10(s, N-H); HDO, 3.57(s, HDO); DMSO-d₆ residual peak, δ 2.50.

Supplementary Figure 11: The peaks were assigned as: MAI, δ 7.49(s, N-H), 2.33(s, N-CH₃); DMSO-d₆ residual peak, δ 2.50.

Supplementary Figure 12: The peaks were assigned as: EAI, δ 7.57(s, N-H); 2.80(q, J = 7.28 Hz, N-CH₂), 1.09(t, J = 7.32 Hz, N-CH₂-CH₃); DMSO-d₆ residual peak, δ 2.50.”

Extended Data Figure 24: what is this assignment based on? Comparing with the liquid-state spectra in Figure 1, the ¹⁵N of FAI resonates at higher frequencies than that of PbI₂.xNH₃. Why is the assignment in the solid state reversed?

Response:

Thank you for your considerate suggestion. Since N atom in FAI is bonded by intramolecular covalent bond, which have the same chemical environment no matter in solution or solid state in the absence of hydrogen bond. And we observed the almost consistent chemical shift values of ¹⁵N signals (-356.33 vs -356.55 ppm) in both states. So we assigned the signal at -356.55 ppm in solid-state ¹⁵N NMR as the ¹⁵N signal in FAI(¹⁵N) without hydrogen bonding. While the interaction between NH₃ and PbI₂ is the intermolecular coordination, which will vary with different phase states. In the solution state, because the solvent DMSO also partially participates in the coordination with PbI₂, the NH₃ donates less electron cloud density towards PbI₂. In contrast, in solid state, without DMSO coordination, NH₃ will donate more electron cloud density towards PbI₂, resulting in its chemical shift value moving to the low field. So, the signal at -343.62 ppm in solid-state ¹⁵N NMR was assigned as ¹⁵N signal in ¹⁵NH₃ coordinated with PbI₂.

Extended Data Figure 19-23: please assign the reference (urea) signal.

Response:

Thank you for the suggestion. We have assigned the reference (urea) signal in the figure caption of Supplementary Fig. 18-22 as follows.

“The ¹⁵N chemical shifts were determined from 1 M urea in DMSO (-303.2 ppm, 10% ¹⁵N labelled) as external standard reference.”

Figure 1: regarding “FAI treated with ¹⁵NH₃ and vacuumed (FAI(¹⁵N))” – the authors attribute the ¹⁵N peak to ¹⁵N-labelled FAI. According to Scheme 1, the reaction of FAI with NH₃ should yield NH₄I and FA⁰, which is then evacuated. Table 1 and Ext. Data Fig. 16 show that the

reaction product is FAI, but this creates confusion (since Scheme 1 states that R can be equal H). Based on Scheme 1 it is also unclear how the ^{15}N -labelled nitrogen of NH_3 would end up in FAI. Figure 1c clarifies this issue but it is still in apparent contradiction to the generality of Scheme 1.

Response:

Thank you for the suggestion. According to Scheme 1, there are two reactions between FAI and NH_3 , one is the ion exchange reaction, and another is transimination reaction. The both reactions occur simultaneously. On one hand, the ion exchange reaction is reversible, and there is a dynamic equilibrium process in the formed liquid phase $\text{FAI}\cdot\text{xNH}_3$ as shown in Supplementary Fig. 16(b). When the liquid phase $\text{FAI}\cdot\text{xNH}_3$ was evacuated, the volatile ammonia (NH_3) in the system will be greatly reduced and the dynamic equilibrium process will be destroyed and the final product is still FAI powder. On the other hand, during the transimination reaction, one external NH_3 molecular added to imine band in FAI, then the other NH_3 was eliminated from reaction intermediate to form the FAI once again (Figure 1c & Scheme 1: R = H). When the external NH_3 is labeled by ^{15}N , it can be seen as a different amine with the unlabeled NH_3 . While the transimination reaction occurring between the FAI and $^{15}\text{NH}_3$, the ^{15}N -labelled FAI will be generated.

Figure 1b also contains a number of peaks which are not assigned (two small peaks in the $\text{FAI}\cdot\text{xNH}_3$ spectrum), suggesting that the reaction is more complex. Why is there no signal from the $^{15}\text{NH}_3$ species in this case while $\text{FAPbI}_3\cdot\text{xNH}_3$ contains it?

Response:

Thank you for your suggestion. In $\text{FAI}\cdot\text{xNH}_3$ sample, there are a variety of hydrogen bond interaction patterns, including N-H \cdots I hydrogen bond and N-H \cdots N hydrogen bond, in which the N-H \cdots N hydrogen bond also includes the hydrogen bond between NH_3 -FA or NH_3 - NH_3 . The small peaks in the spectrum are the signals of NH_3 under different hydrogen bond patterns. These hydrogen bond patterns are complex and difficult to assign one by one. Here, we only assigned the main signals with strong and relatively fixed chemical shift values in different spectra. In $\text{FAPbI}_3\cdot\text{xNH}_3$ spectrum, the signal at -259.47 ppm is supported as the $^{15}\text{NH}_3$ signal, which has both coordination interaction and hydrogen bond interaction. Corresponding modifications have been revised in manuscript and supporting information.

“The hydrolysis reactions of FA^0 , RFA^0 and DRFA^0 lead to the formation of volatile HCOOH and RA^0 ” – what is this conclusion based on? How was the presence of the volatile products concluded?

Response:

Thank you for the suggestion. The hydrolysis reactions of FA^0 , RFA^0 and DRFA^0 leading to the formation of volatile HCOOH and RA^0 have been reported in 1890s (Ref: *Justus Liebigs Annalen der Chemie*, 1895, 286, 343-368, doi: 10.1002/jlac.18952860309; *Chemical Reviews*, 1944, 35, 351-425, doi: 10.1021/cr60112a002, page 381.). Based on the results of experiment Table1 No. 9-11, there were only RAI product remaining. We supported that FA

should degrade to volatile products, and considering the presence of water, we believed the hydrolysis reactions occurring. We have added the above reference in the manuscript.

“The s-triazine with a boiling point of 114 °C is attributed to the transimination reaction of three FA⁰ molecules along with the formation of NH₃” – this is not a transimination reaction but condensation: please refer to <https://pubs.acs.org/doi/abs/10.1021/acs.jpcc.7b09805> and the references within (in particular by Schaeffer et al., ref. 20).

Response:

Thank you for the suggestion. We have noticed that the reaction mechanism of three FA⁰ molecules to form s-triazine has been mentioned in reference (<https://pubs.acs.org/doi/abs/10.1021/acs.jpcc.7b09805> and Schaeffer et al., ref. 20), and we have added these two references in the main text. According to the possible mechanism, this condensation still follows the addition-elimination pathway as the same as transimination reaction, and here, R group in R-A (Scheme 1) is an imine group (-CH=NH). So, we support that this condensation belongs to transimination reaction.

Finally, since NMR is not a commonly used technique in the field of halide perovskite fabrication, it would be beneficial to include a brief introduction to its main tenets and differences relative to XRD etc. A few relevant recent reviews on the subject include <https://www.nature.com/articles/s41570-021-00309-x>

<https://pubs.acs.org/doi/abs/10.1021/jacs.0c07338>

Response:

Thank you for your kind suggestion. A brief introduction about Nuclear magnetic resonance (NMR) spectroscopy has been added in manuscript as follows.

“Nuclear magnetic resonance (NMR) spectroscopy is a particularly practical tool for quantifying the relative amounts of the organic cations and related chemical reactions in the perovskite precursor.”

REVIEWER COMMENTS

Reviewer #2 (Remarks to the Author):

1. The TRPL data look abnormal to me. The bare perovskite films without charge extraction layer should exhibit mono-exponential decay profile, where the trap-assisted recombination dominates. However, the raw perovskite film clearly displays a bi-exponential characteristic, can you explain?

2. Both the the raw and post-treated perovskite films were fitted into a bi-exponential decay. Can you explain why fit into bi-exponential? What are the physical meanings of the fast and slow components?

3. The lifetime were only 132 ns for the treated films, which is much shorter than the previous studies for FACs based perovskites (typically a few hundreds of nanoseconds to microseconds). Any explanation?

4. I do not see experimental details of the TRPL measurement in the manuscript, which should be added.

Reviewer #4 (Remarks to the Author):

The authors have now addressed my queries in full. The results and their analysis are of high quality and warrant publication.

One final minor comment: there is an issue with the reference numbers in the revised manuscript, please double check.

"Nuclear magnetic resonance (NMR) spectroscopy is a particularly practical tool for quantifying the relative amounts of the organic cations and related chemical reactions in the perovskite precursor"

The authors accidentally reference in this sentence the works related to FA degradation (30-33) whereas the pertinent works would be general review articles providing an overview of the use of NMR in halide perovskite studies, for example:

<https://www.nature.com/articles/s41570-021-00309-x>

<https://pubs.rsc.org/en/content/articlelanding/2021/ta/d1ta03572j>

Reviewer #2 (Remarks to the Author):

1. The TRPL data look abnormal to me. The bare perovskite films without charge extraction layer should exhibit mono-exponential decay profile, where the trap-assisted recombination dominates. However, the raw perovskite film clearly displays a bi-exponential characteristic, can you explain?
2. Both the raw and post-treated perovskite films were fitted into a bi-exponential decay. Can you explain why fit into bi-exponential? What are the physical meanings of the fast and slow components?

Response: Thank you for your suggestion. As for the pure perovskite films, there are generally three mechanisms determine PL decay: first-order (trap-assisted nonradiative); second-order (radiative); and third-order (Auger) recombination, and their contributions depend on the intensity of the excitation laser fluence (*Nat. Commun.* 9, 3021 (2018); *Nat. Energy* 6, 63 (2021)). The third-order recombination constant under this carrier density is very small ($k_3 \ll 10^{-30} \text{ cm}^6/\text{s}$) and is only narrowly observed in the initial stage (*Nat. Energy* 6, 63-71 (2021)). Thus, the third-order mechanism in PL decay analysis is usually neglected. Commonly, the carrier lifetime values were obtained by using the biexponential equation $Y = A_1 \exp(-t/\tau_1) + A_2 \exp(-t/\tau_2)$, where τ_1 and τ_2 denote the fast and slow decay time and are related to the trap-assisted nonradiative and radiative recombination processes, respectively. (*Science* 370, 108-112 (2020); *Nat. Commun.* 12, 1554 (2021)).

3. The lifetime were only 132 ns for the treated films, which is much shorter than the previous studies for FACs based perovskites (typically a few hundreds of nanoseconds to microseconds). Any explanation?

Response: Thank you for your helpful suggestion. The lifetime of 132.78 ns and 10.06 ns in the manuscript is the fast components which is related to the trap-assisted nonradiative, and the longer time in the previous studies for FACs based perovskites generally refer to the average lifetime τ_{ave} (*Nat. Commun.* 12, 6603 (2021); *Nat. Commun.* 12, 1554 (2021)). To avoid ambiguity, we have revised the expression in the manuscript. We also reprocessed the TRPL results in Supplementary Figure 30c and the lifetime of the trap-assisted nonradiative (τ_1) was fitted into 6.50 ns and 58.45 ns for Raw and $\text{NH}_3\text{-FACsPbI}_3$ films, respectively. The value of fitting depends heavily on the selection range of data, and our data is equivalent to the results of *Nat. Commun.* 9, 3021 (2018). The relevant revision is as follows:

"The reduced defect density is also evidence by the charge-limited current (SCLC) and time-resolved photoluminescence (TRPL) results in Supplementary Fig. 30. The calculated defect density from the SCLC curves decreases from 3.97×10^{16} to $9.62 \times 10^{15} \text{ cm}^{-3}$. And the TRPL results shows that the $\text{NH}_3\text{-FACsPbI}_3$ film has a much longer trap-assisted nonradiative lifetime ($\tau_1=58.45 \text{ ns}$) than the Raw FACsPbI_3 film ($\tau_1=6.50 \text{ ns}$)."

Supplementary Figure 30 | (c) Time-resolved photoluminescence decay curves of the Raw and $\text{NH}_3\text{-FACsPbI}_3$ films. The carrier lifetime values were fitted by using the biexponential equation $Y = A_1\exp(-t/\tau_1) + A_2\exp(-t/\tau_2)$, where τ_1 and τ_2 denote the fast and slow decay time and are related to the trap-assisted nonradiative and radiative recombination processes, respectively. The films were photoexcited at 405 nm, and emission were collected on the surface side of the film (perovskite/FTO glass substrate).

4. I do not see experimental details of the TRPL measurement in the manuscript, which should be added.

Response: Thank you for your considerable suggestion, we have added the experimental details of the TRPL measurement to the section of Perovskite film and Device Characterization. The relevant revision is as follows:

“Time-resolved photoluminescence (TRPL) experiments were performed with a Fluorolog using a pulsed source at 405 nm (NanoLED 402-LH from Horiba, pulse width <200 ps, 11 pJ/pulse, approx. 1 mm² spot size), and the signal was recorded at 805 nm using the Time-Correlated Single Photon Counting (TCSPC) technique.”

Reviewer #4 (Remarks to the Author):

The authors have now addressed my queries in full. The results and their analysis are of high quality and warrant publication.

One final minor comment: there is an issue with the reference numbers in the revised manuscript, please double check.

"Nuclear magnetic resonance (NMR) spectroscopy is a particularly practical tool for quantifying the relative amounts of the organic cations and related chemical reactions in the perovskite precursor"

The authors accidentally reference in this sentence the works related to FA degradation (30-33) whereas the pertinent works would be general review articles providing an overview of the use of NMR in halide perovskite studies, for example:

<https://www.nature.com/articles/s41570-021-00309-x>

<https://pubs.rsc.org/en/content/articlelanding/2021/ta/d1ta03572j>

Response: Thank you for your helpful suggestion, we have replaced the corresponding reference in the manuscript. The relevant revision is as follows:

“(31) Dahلمان, C. J.; Kubicki, D. J.; Reddy, G. N. M., Interfaces in metal halide perovskites probed by solid-state NMR spectroscopy. *J. Mater. Chem. A* **9**, 19206-19244 (2021).

(32) Kubicki, D. J.; Stranks, S. D.; Grey, C. P.; Emsley, L., NMR spectroscopy probes microstructure, dynamics and doping of metal halide perovskites. *Nat. Rev. Chem.* **5**, 624–645 (2021).”

REVIEWER COMMENTS

Reviewer #2 (Remarks to the Author):

1. Regarding the interpretation of TRPL data there is nothing to do with Auger, because the excitation laser fluence is too low for the Auger to kick in. I would also be highly suspicious if the bimolecular recombination would happen at such a low excitation intensity. I would expect only monomolecular recombination occurred in the film. The authors attribute the fast and slow decays of the TRPL to trap-assisted nonradiative and radiative process, respectively, which should be wrong in this circumstance.

2. Once again, the PL lifetime is only around 100 ns which usually observed in defective films, but seems to be unusual for the high efficiency perovskite films as the authors reported. I suggest the authors to remeasure the TRPL data to double check if the measurements were properly carried out.

Reviewer #2 (Remarks to the Author):

1. Regarding the interpretation of TRPL data there is nothing to do with Auger, because the excitation laser fluence is too low for the Auger to kick in. I would also be highly suspicious if the bimolecular recombination would happen at such a low excitation intensity. I would expect only monomolecular recombination occurred in the film. The authors attribute the fast and slow decays of the TRPL to trap-assisted nonradiative and radiative process, respectively, which should be wrong in this circumstance.

2. Once again, the PL lifetime is only around 100 ns which usually observed in defective films, but seems to be unusual for the high efficiency perovskite films as the authors reported. I suggest the authors to remeasure the TRPL data to double check if the measurements were properly carried out.

Response: Thank you for your kind suggestion. The data in the old version may be insufficient due to the decomposition of the sample during mailing process. We remeasured the TRPL data and relevant TRPL results in Supplementary Figure 30c have been revised. The lifetime of the Raw and NH_3 -FACsPbI₃ films was fitted into 20.49 μs and 41.34 μs , respectively.

The relevant revision is as follows:

“The reduced defect density is also evidence by the charge-limited current (SCLC) and time-resolved photoluminescence (TRPL) results in Supplementary Fig. 30. The calculated defect density from the SCLC curves decreases from 3.97×10^{16} to $9.62 \times 10^{15} \text{ cm}^{-3}$. And the TRPL results show that the NH_3 -FACsPbI₃ film has a much longer lifetime ($\tau=41.34 \mu\text{s}$) than the Raw FACsPbI₃ film ($\tau=20.49 \mu\text{s}$).”

Supplementary Figure 30 | (c) Time-resolved photoluminescence decay curves of the Raw and NH_3 -FACsPbI₃ films.

Reviewers' comments:

Reviewer #2 (Remarks to the Author):

How are the lifetimes extracted from the TRPL? It seems the lifetimes are determined simply by the time to ground state which is wrong. Besides, to my knowledge, I have never seen lifetimes of tens of microseconds reported. If the lifetime is actually that long, the defect density in the films should be very low. However, from the SCLC, the defect density is around 10^{16} which is very high. The results are not coherent.

Dear reviewer 2#,

Q: How are the lifetimes extracted from the TRPL? It seems the lifetimes are determined simply by the time to ground state which is wrong. Besides, to my knowledge, I have never seen lifetimes of tens of microseconds reported. If the lifetime is actually that long, the defect density in the films should be very low. However, from the SCLC, the defect density is around 10^{16} which is very high. The results are not coherent.

A: Thanks for your comments. We further checked and recalculated the lifetimes of TRPL, finding an error for the fitting value in the last version. We are very sorry about it. Now we corrected the lifetimes that are 3.2 and 5.6 μs for the Raw FACsPbI₃ and NH₃-FACsPbI₃, in the revised manuscript, respectively. We are really grateful for your comment and reminding. We hope that the revised version can meet the strict requirements and high standards of Nature Communications.

Best,

Shuping Pang

Further Editorial Questions:

- what were the previous errors you found regarding TRPL/SCLC/lifetimes;
- how were the errors addressed;
- step-by-step methodological process from acquiring the data, processing, analysis, and the calculations performed.

Response:

Thanks for giving us an opportunity to explain our errors in the last revised manuscript.

There are two errors. One is that we used the bi-exponential fitting instead of the mono-exponential fitting, and the other is the error in unit conversion.

In the last revised manuscript, the wrong statement is “the TRPL results show that the NH₃-FACsPbI₃ film has a much longer lifetime ($\tau=41.34 \mu\text{s}$) than the Raw FACsPbI₃ film ($\tau=20.49 \mu\text{s}$)”. The lifetime was calculated by bi-exponential fitting with the average lifetimes are of 4314 ns and 2049 ns for the perovskite films after and before NH₃ healing. Therefore, the lifetime should be 4.134 μs and 2.049 μs , but not 41.34 μs and 20.49 μs . The error happened in the unit conversion.

Former processing:

The TRPL original data is mapped by Origin software, and fitted with the Expdec2

function ($y = y_0 + A_1e^{-x/t_1} + A_2e^{-x/t_2}$) as demonstrated in Figure R1. We calculated the average value through the formula of $\tau_{ave} = \frac{\sum A_i \tau_i^2}{\sum A_i \tau_i}$, which A_i refers to the fitting factor of each stage of the curve and τ_i refers to the transient fluorescence lifetime value of each stage.

Figure R1. Screenshot of data processing using Origin software fitted according to the Expdec2 function ($y = y_0 + A_1e^{-x/t_1} + A_2e^{-x/t_2}$)

As the reviewer suggested before, the lifetime should be calculated by mono-exponential fitting of the perovskite film without charge extraction layer (the previous comment is as follows).

Reviewer #2 (Remarks to the Author):

1. The TRPL data look abnormal to me. The bare perovskite films without charge extraction layer should exhibit mono-exponential decay profile, where the trap-assisted recombination dominates. However, the raw perovskite film clearly displays a bi-exponential characteristic, can you explain?

Here, the same curves have been calculated as the reviewer suggested by the mono-exponential fitting ($y = y_0 + A_1e^{-x/t_1}$) as demonstrated in Figure R2, and the lifetimes value are 3.2 and 5.6 μ s for the Raw FACsPbI₃ and NH₃-FACsPbI₃ films, respectively.

Figure R2. Screenshot of data processing using Origin software fitted according to the ExpDec1 function ($y = y_0 + A_1 e^{-x/t_1}$)

REVIEWERS' COMMENTS

Reviewer #2 (Remarks to the Author):

I thank the authors for the clarification. The control (untreated) film shows quite a long PL lifetime of over 2 microseconds. The control films are full of pinholes and vacancies, indicating its poor quality. Although the morphology does not necessarily indicate strong PL quenching, 2 microseconds is still too long for such a defective film. FYI, the perovskite used in the champion inverted cells published recently on Science is only around 2 microseconds (Science 376, 416–420 (2022)). The authors need to explain why such a defective film show very long PL lifetime. Also, I strongly suggest the authors to double check the TRPL to verify if the lifetimes are consistent in different samples.

For the small-area cell, the authors provided certified efficiency, however, the certified efficiency is 1% lower than the lab-measured one. If the discrepancy still exists or even amplified for large-area modules remain uncertain. Considering the champion efficiency to date have achieved 25.7% for small-area nip structured cells, the 22% certified efficiency reported in this work is not very high. In this sense, the large-area modules would be critical important to justify the importance of the work. I recommend to include certified efficiency of the fabricated modules in the paper.

Reviewer #2 remarks:

I thank the authors for the clarification. The control (untreated) film shows quite a long PL lifetime of over 2 microseconds. The control films are full of pinholes and vacancies, indicating its poor quality. Although the morphology does not necessarily indicate strong PL quenching, 2 microseconds is still too long for such a defective film. FYI, the perovskite used in the champion inverted cells published recently on Science is only around 2 microseconds (Science 376, 416–420 (2022)). The authors need to explain why such a defective film show very long PL lifetime. Also, I strongly suggest the authors to double check the TRPL to verify if the lifetimes are consistent in different samples.

We thank the reviewer for all the helpful comments and suggestions.

Response:

We remeasured the lifetimes of the $\text{NH}_3\text{-FACsPbI}_3$ film and the Raw-FACsPbI₃ film based on time-resolved photoluminescence (TRPL), as shown in Figure R1. In order to avoid the effect from humidity, a raw-FACsPbI₃ film with polymethyl methacrylate (PMMA) protective layer is also added. The PL lifetime of $\text{NH}_3\text{-FACsPbI}_3$ is 4.0 μs , and the PL lifetimes of Raw-FACsPbI₃ without PMMA and with PMMA are 2.4 and 2.1 μs , respectively. The remeasured PL lifetime results are consistent with those of the last time. The excitation light of the films is at 483.6 nm with pulse width of ~ 118.6 ps and power of 5 mW/pulse, and the emission light is collected from the side of perovskite surface for the film (Edinburgh FLS980). The pinholes in the raw film leading to low efficiency of the device is mainly attributed to the connection of hole transport layer (HTL) and electrode transport layer (ETL), which leads to the serious carrier nonradiative recombination. The lifetimes from TRPL spectra for perovskite films (perovskite/ glass substrate) are closely related with the vacancy and nonradiative recombination in perovskite films, rather than the interfaces of perovskite/ETL, perovskite/HTL or ETL/HTL.

We also noted that the measuring conditions such as the wavelength and intensity of the excitation light show obvious effect on lifetime (Energy Environ. Sci., 2022,15, 2096-2107), which may lead to the large difference with similar perovskite films from different groups. The preparation details of thin film samples are provided in the notes of Supplementary Fig. 32.

Figure R1 Time-resolved photoluminescence decay curves of the Raw- and $\text{NH}_3\text{-FACsPbI}_3$ films.

For the small-area cell, the authors provided certified efficiency, however, the certified efficiency is 1% lower than the lab-measured one. If the discrepancy still exists or even amplified for large-area modules remain uncertain. Considering the champion efficiency to date have achieved 25.7% for small-area nip structured cells, the 22% certified efficiency reported in this work is not very high. In this sense, the large-area modules would be critical important to justify the importance of the work. I recommend to include certified efficiency of the fabricated modules in the paper.

Response: We certified the modules and obtained a certified efficiency of 19.38% on an active area of 14 cm². The corresponding report is provided in the supporting information (Supplementary Fig. 38).

[REDACTED]

Supplementary Fig. 38 Certified efficiency of the module based on the NH₃-FACsPbI₃ film from Photovoltaic and Wind Power Systems Quality Test Center, IEE, CAS, China. No. PWQC-WT-P21121521-1R.